**Technical Report**

# BridgePRS leverages shared genetic effects across ancestries to increase polygenic risk score portability

Clive J. Hoggart [1] ✉, Shing Wan Choi [1,2], Judit García-González [1], Tade Souaiaia[3], Michael Preuss [4] & Paul F. O'Reilly [1] ✉

Here we present BridgePRS, a novel Bayesian polygenic risk score (PRS) method that leverages shared genetic effects across ancestries to increase PRS portability. We evaluate BridgePRS via simulations and real UK Biobank data across 19 traits in individuals of African, South Asian and East Asian ancestry, using both UK Biobank and Biobank Japan genome-wide association study summary statistics; out-of-cohort validation is performed in the Mount Sinai (New York) Bio*Me* biobank. BridgePRS is compared with the leading alternative, PRS-CSx, and two other PRS methods. Simulations suggest that the performance of BridgePRS relative to PRS-CSx increases as uncertainty increases: with lower trait heritability, higher polygenicity and greater between-population genetic diversity; and when causal variants are not present in the data. In real data, BridgePRS has a 61% larger average $R^2$ than PRS-CSx in out-of-cohort prediction of African ancestry samples in Bio*Me* ($P = 6 \times 10^{-5}$). BridgePRS is a computationally efficient, user-friendly and powerful approach for PRS analyses in non-European ancestries.

PRSs have typically been derived using European ancestry genome-wide association study (GWAS) data, resulting in substantially lower predictive power when applied to non-European samples, in particular those of African ancestry[1,2]. The PRS trans-ancestry portability problem is well established and is caused by marked differences in linkage disequilibrium (LD), differences in allele frequency driven by genetic drift and natural selection, and gene–environment interactions affecting causal effect sizes[3]. Consequently, the etiological insights and clinical utility provided by PRS derived in Europeans may have limited relevance to individuals of non-European ancestries.

Increasing GWAS sample sizes for underrepresented populations is of critical importance for improving their PRS. However, optimal power will be achieved by using all GWASs available across ancestries for PRS prediction in any one ancestry; this is because causal genetic effect sizes are highly correlated globally, even between genetically distant ancestries[4]. PRS-CSx[5], developed to tackle the PRS portability

problem, makes cross-population inference on the inclusion of each single-nucleotide polymorphism (SNP) across the genome (or, more precisely, the degree of shrinkage of variant effect sizes to zero). PRS-CSx uses Bayesian modeling with a prior that strongly shrinks small effect sizes to zero, reducing the number of candidate SNPs to a minimal set. This is analogous to fine-mapping of causal variants. However, although the inclusion of causal variants in the PRS is ideal, fine-mapping approaches may not be as effective when causal variants are missing or when power is insufficient for them to be accurately identified.

We introduce BridgePRS, a novel Bayesian PRS method that also integrates trans-ancestry GWAS summary statistics. Unlike the fine-mapping approach of PRS-CSx, BridgePRS retains all variants within loci to best tag causal variants shared across ancestries. The focus is on correctly estimating causal effect sizes, which is key when the goal is prediction, rather than on estimating their location. This

[1]Department of Genetics and Genomic Sciences, Icahn School of Medicine, Mount Sinai, New York, NY, USA. [2]Regeneron Genetics Center, Tarrytown, NY, USA. [3]Department of Cellular Biology, Suny Downstate Health Sciences, Brooklyn, NY, USA. [4]The Charles Bronfman Institute for Personalized Medicine, Icahn School of Medicine, Mount Sinai, New York, NY, USA. ✉e-mail: clivehoggart@gmail.com; paul.oreilly@mssm.edu

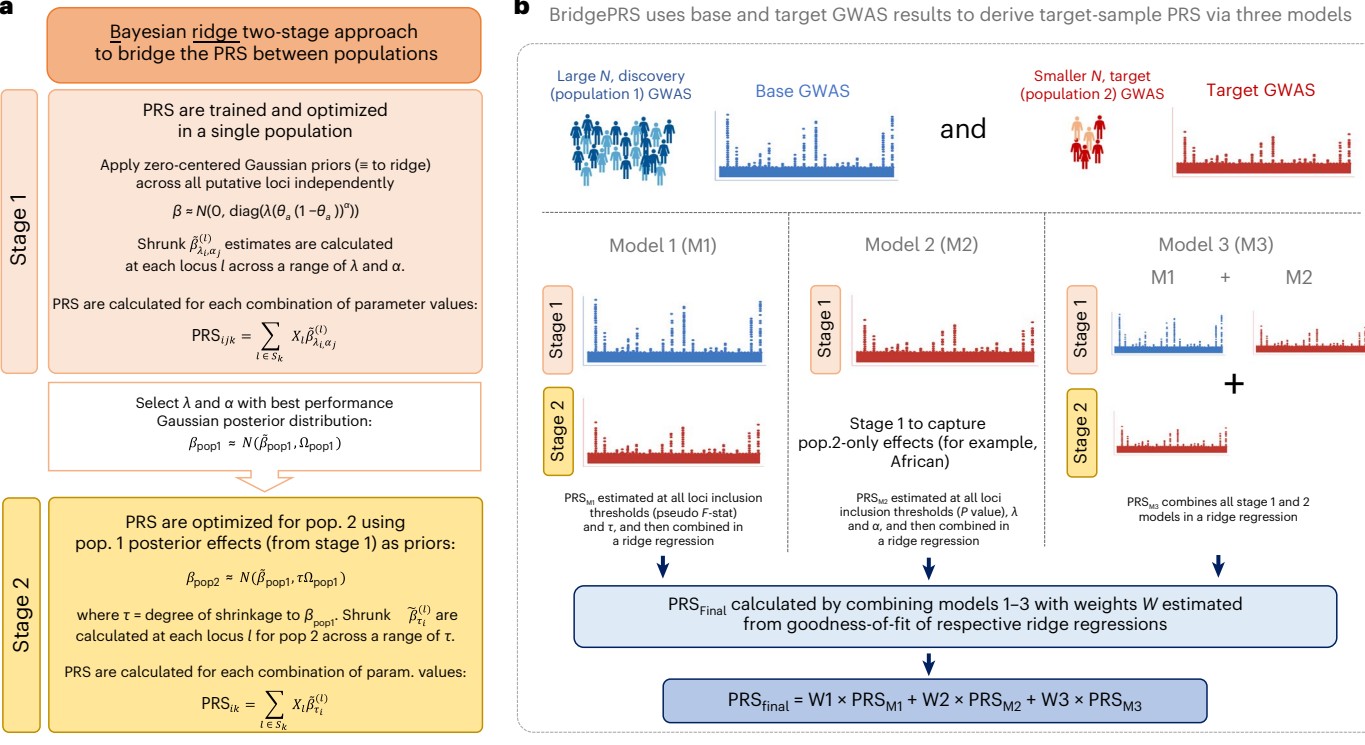

**Fig. 1 | Flow diagram describing the modeling of BridgePRS. a**, Two-stage approach to combine GWASs from two populations. **b**, BridgePRS combining three different PRS models to determine the final PRS. $\beta$, SNP effect sizes; $\lambda$, shrinkage coefficients; $\theta_a$, allele frequency of SNP $a$; $\alpha$, parameter for dependency between effect size and allele frequency; $\tau$, degree of shrinkage of population 2 effects, $\beta_{pop2}$, to those of population 1, $\beta_{pop1}$; $X_l$, genotypes at locus $l$, where a locus is defined as a region in which SNPs are correlated ($r^2 > 0.01$) with each other; $\tilde{\beta}^{(l)}$, posterior mean SNP effects at locus $l$, where subscripts denote prior parameters used; $\Omega_{pop1}$, posterior precision matrix for population 1 using the best-fitting

prior parameters $\alpha$ and $\lambda$ (the Gaussian distribution is parameterized by its precision matrix (inverse covariance matrix), throughout). $S_k$ is the set of loci whose rank exceeds a threshold of $k$: in stage 1 loci are ranked by the $P$ value of their top SNP, whereas in stage 2 loci are ranked by the pseudo $F$ statistic, which measures the joint association of all SNPs at the locus in the target population; $i$ and $j$ index over prior parameters; and $W$ are the weights obtained from goodness-of-fit of the best-fitting ridge regression model that combines models 1–3. This figure simplifies the modeling for brevity (see Methods for details).

approach is less reliant on the inclusion and identification of causal variants. BridgePRS is most applicable to combining the information of a well-powered GWAS performed in a (discovery) population or populations not matched to the ancestry of the target sample, with a second GWAS of relatively limited power in a (target) population that is well-matched to the ancestry of the target sample.

We apply BridgePRS to simulated data and compare its performance with that of PRS-CSx and two single-ancestry PRS methods adapted to use GWAS data from multiple ancestries. The simulations demonstrate the different scenarios in which BridgePRS and PRS-CSx are optimal. We then use UK Biobank (UKB)[6] and Biobank Japan (BBJ)[7,8] GWAS data to construct PRS for African, South Asian and East Asian ancestry samples. The resultant PRSs are validated in unseen UKB samples and in the entirely independent New York-based Mount Sinai Bio*Me* biobank[9], producing results consistent with the simulations.

## Results

### Overview of BridgePRS method

An overview of the BridgePRS modeling employed here is shown in Fig. 1. The key modeling (model 1 in Fig. 1; Methods) can be broken into two stages: (1) a PRS is trained and optimized using data from a large discovery population (for example, European) GWAS, with a zero-centered Gaussian prior distribution for SNP effect sizes (analogous to ridge regression) within putative loci; and (2) the SNP effect sizes of this PRS are treated as priors and updated in a Bayesian framework by those of the smaller target population (for example, African) GWAS. Thus, the

two-stage Bayesian ridge approach of BridgePRS 'bridges' the PRS between the two populations.

The main causes of poor trans-ancestry PRS portability are differences in LD and allele frequencies between populations[3]. Differences in LD result in the best tag for a causal variant differing between populations. To account for the resultant uncertainty in the location of causal variants, BridgePRS averages SNP effects across SNPs within putative loci instead of selecting a single best SNP as performed by standard clumping and thresholding (C+T) PRS[10]. BridgePRS is first applied to the discovery population GWAS, using Bayesian modeling with zero-centered Gaussian priors, equivalent to penalized likelihood ridge regression, at putative loci. Given summary data from large GWAS in Europeans, we find that this procedure alone improves predictive accuracy in African and South Asian target data compared with choosing single best SNPs at putative loci. Thus, whereas the main BridgePRS method uses GWAS data from the discovery and target GWAS, the option of using only discovery GWAS is available in the BridgePRS software.

Stage 1 modeling results in multivariate Gaussian posterior distributions for SNP effect sizes at each locus. Stage 2 modeling integrates the (smaller) target population GWAS data into the PRS by using this posterior distribution as a prior distribution for SNP effect sizes of the target population. Stage 2 allows for different effect size estimates between the populations, caused by differences in LD, in allele frequencies driven by drift or selection, and by differences in causal effect sizes due to gene–environment interactions. Stages 1 and 2 both use conjugate prior–posterior updates, providing computationally

efficient analytical solutions and enabling BridgePRS analyses to be performed rapidly.

Variation in causal allele frequencies between populations can mean that causal variants with relatively low minor allele frequency in the discovery population are estimated with large errors or missed altogether. To ameliorate this problem, PRSs are derived by applying BridgePRS stage 1 modeling to the target population data alone (model 2 in Fig. 1; Methods). Model 1 and model 2 PRSs are combined in model 3 (Fig. 1 and Methods).

Each stage of the modeling is fit across a spectrum of prior parameters and criteria to select loci for inclusion in the PRS calculation, with each combination of parameters giving rise to a unique PRS. These PRSs are then combined in a ridge regression fit using available genotype–phenotype test data, choosing the optimal ridge penalty parameters by cross-validation (Methods).

### Benchmarking methods via simulation

We used the HAPGEN2 software[11] to simulate HAPMAP3 variants for 100,000 European, 40,000 African and 40,000 East Asian ancestry samples using 1000 Genomes Phase 3 (1000G) samples[12] as a reference. Simulations were restricted to 1,295,289 variants with minor allele frequency >1% in at least one of the three populations. Phenotypes were subsequently simulated under three models of genetic architecture in which causal variants were sampled from 1%, 5% and 10% of the available HAPMAP3 variants. Population-specific effect sizes were sampled from a multivariate Gaussian distribution with between-population correlation of 0.9. Genetic effects were combined assuming additivity, and Gaussian noise at two levels of variance was added to generate phenotypes with 25% and 50% SNP heritability. For each of the six scenarios of polygenicity and heritability, ten independent phenotypes were generated and analyses were run with and without inclusion of the causal variants.

Data were split into training for GWAS (80,000 European, 20,000 non-European), with the remainder split equally into 10,000 samples for model optimization (test data) and assessment of model performance (validation data). The performance of BridgePRS was compared with that of PRS-CSx, PRS-CS-mult and PRSice-meta. PRS-CS-mult applies the single-ancestry PRS-CS method[13] to the populations under study and combines them by estimating weights in a linear regression using the test data. PRSice-meta applies clumping and thresholding, as implemented in PRSice[14], to meta-analysis of the populations under study, selecting the LD panel from the two populations under study that optimizes prediction in the test data of the target population.

Polygenicity ranging from 1% to 10% (fraction of variants with nonzero effect sizes) is consistent with the findings of a recent study of 28 complex traits in the UKB[15]. Between-population correlation of causal variant effect sizes of 0.9 is consistent with the results of a multiancestry lipids GWAS in which causal variants were fine-mapped[16] and with a recent study estimating a mean genetic correlation of 0.98 of causal variant effect sizes between ancestries across a range of continuous traits[4]. Approximately one-third to two-thirds of heritability is captured by common SNPs[17]; therefore, our simulation at 25% heritability implies a total heritability of 37.5–75.0%. The power of GWAS, and therefore PRS, is a function of sample size and heritability, such that doubling heritability is equivalent to doubling sample size in terms of power, as the standard error of a GWAS regression coefficient is the same if either the sample size or heritability is doubled (Methods). Therefore, our simulations at 50% SNP heritability and GWAS with 80,000 European samples are equivalent to 25% SNP heritability and GWAS with 160,000 European samples.

Figure 2 summarizes the results from PRS analyses performed on simulated data. Both BridgePRS and PRS-CSx outperformed the single-ancestry methods across all scenarios. BridgePRS performed better than PRS-CSx in analyses of African samples with 5% and 10% of variants assigned as causal. With 1% of variants causal, the methods

had similar accuracy when causal variants were not included and at 25% heritability, and PRS-CSx performed better with causal variants included at 50% heritability. In analyses of East Asian samples, the same relative pattern was observed, but the differences were less pronounced, and PRS-CSx performed better in all scenarios in which 1% of variants were causal. Across the analyses, BridgePRS performed better compared with PRS-CSx when the causal variants were not included in the data (Extended Data Fig. 1). Overall, the simulations reveal that the performance of BridgePRS relative to that of PRS-CSx increases as the uncertainty increases: at lower heritability, higher polygenicity, greater between-population genetic diversity and when causal variants are not present in the data.

The theoretical proportion of heritability ($h^2$) captured by a PRS derived by C+T, assuming independent causal variants, is $r^2/h^2 = (1 + m/nh^2)^{-1}$, where $r^2$ is the variance explained by the PRS, $m$ is the number of causal variants and $n$ is the GWAS sample size[18,19]. Although BridgePRS and PRS-CSx are more sophisticated methods than C+T, the factor $nh^2/m$ in the equation, which is a measure of power to detect individual causal variant effects, is useful in describing the relative performance of the methods. Figure 2 shows results in relation to $nh^2/m$ (up to a proportionality constant): lower values favor Bridge-PRS, higher values favor PRS-CSx, and within the same target population the relative performance of the methods is similar for constant $nh^2/m$. For example, results at 25% heritability and 5% causal variants showed the same relative method performance as results at 50% heritability and 10% causal variants, for both African and East Asian target samples (Fig. 2a versus Fig. 2b), as expected.

Extended Data Fig. 2 shows results for the same simulation settings as those used in the main analysis (Fig. 2) but with the GWAS training sample size halved (40,000 European, 10,000 non-European). Here, the performance of BridgePRS relative to PRS-CSx increased compared with the results with the full GWAS samples sizes at 50% heritability, and as predicted, the relative performance of the methods at 50% heritability was similar to that at 25% heritability and the full GWAS sample sizes. Extended Data Fig. 3 shows results at the original GWAS sample size and 75% heritability (equivalent to 240,000 European, 60,000 non-European GWAS training sample sizes and 25% heritability). As predicted, the performance of BridgePRS relative to PRS-CSx decreased compared with the results at 25% and 50% heritability.

These simulation analyses used 1000G data as their reference LD panel, that is, the correct LD panel. To assess the sensitivity of the methods to misspecification of LD, analyses were rerun using UKB data to estimate ancestry-specific LD. Extended Data Fig. 4 shows the performance of BridgePRS and PRS-CSx using an LD reference panel constructed from African and East Asian UKB samples relative to their performance using the 1000G reference panel. Both methods exhibited a minimal loss in predictive accuracy using UKB reference panels.

### Benchmarking methods via real data

The four PRS methods were applied to UKB[6] samples of African and South Asian ancestry across 19 continuous anthropometric and blood measure traits (for East Asian ancestry, see below). These traits were selected to maximize heritability and samples sizes of non-European individuals and to minimize their pairwise correlation (maximum $r^2 < 0.3$; Methods). For each trait, UKB samples of European, African and South Asian ancestry were split into training, test and validation sets in proportions of 2/3, 1/6 and 1/6, respectively. Sample sizes are shown in Extended Data Table 1. The training data were used to generate GWAS summary statistics, and the test data were used to select optimal model parameters. Results are shown for the resultant PRS in the unseen UKB validation data. In addition, an entirely out-of-sample validation study was performed by applying the PRS derived in the UKB to BioMe[9] for the nine traits also available in BioMe.

Within the UKB there were 2,472 East Asian samples, which was too few to split into training (GWAS), test and validation sets as above.

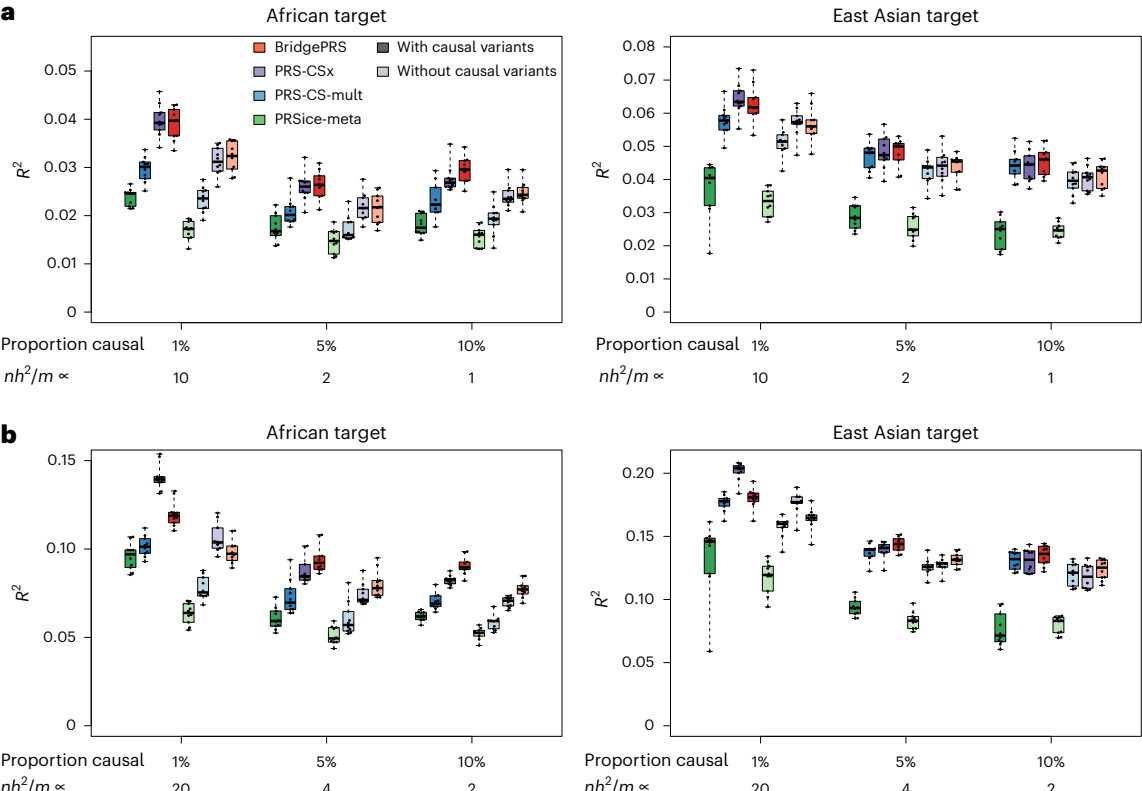

**Fig. 2 | Predictive accuracy for different polygenic prediction methods in simulations.** Results are shown for BridgePRS, PRS-CSx, PRS-CS-mult and PRSice-meta across six simulation scenarios, with and without the causal variants included in the model for African and East Asian ancestry target samples. **a,b**, SNP heritability $h^2_{snp} = 0.25$ (**a**) and SNP heritability $h^2_{snp} = 0.5$ (**b**); ten simulated phenotypes per scenario. Under each set of analyses, the proportion of causal variants and the relative power of the data used are shown, measured by $nh^2/m$ up to proportionality, where $n$ is the GWAS sample size, $h^2$ is the heritability and $m$ is the number of causal variants. The central rectangular boxes show the interquartile range, horizontal lines inside the boxes show the median, whiskers extend to the most extreme results and points show results for each of the ten simulated phenotypes.

However, GWAS summary statistic data from BBJ were available for download[7,8]. We combined these data with the European UKB GWAS summary statistics described above for 13 overlapping traits to estimate PRS for East Asian ancestry (as above). BridgePRS combines SNP effect size estimates across GWAS (as does the PRSice-meta method) and therefore requires effect sizes to be on the same scale. However, the BBJ summary statistics were generated after standardizing the trait values to have a mean of zero and a standard deviation of one, whereas the UKB GWASs were applied to raw trait data. Therefore, before applying the methods, the BBJ effect estimates and standard errors were transformed to the respective scale of the UKB measures, assuming that the BBJ and UKB trait values had the same variance. UKB East Asian samples were then split equally into test data for model optimization and validation data to assess model performance, as above. PRSs were also validated in East Asian Bio*Me* samples across eight overlapping traits.

Trait sample sizes for each ancestral population in the UKB and Bio*Me* cohorts are shown in Extended Data Tables 1 and 2. For all analyses, imputed genotype data were used.

Figure 3 shows boxplots of the variance explained ($R^2$) by Bridge-PRS, PRS-CSx, PRS-CS-mult and PRSice-meta, for all traits analyzed, for prediction of African, South Asian and East Asian ancestry samples in the UKB and Bio*Me* cohorts. Also shown are $P$ values comparing the differences in within trait $R^2$, summed across all traits, between Bridge-PRS, PRS-CSx and PRS-CS-mult (not PRSice-meta as it was universally inferior across all comparisons). For prediction of African ancestry samples, BridgePRS had the highest median $R^2$ in UKB (0.031 versus 0.025) and a 61% higher median $R^2$ than PRS-CSx (0.044 versus 0.027)

in the out-of-cohort Bio*Me* samples ($P = 6 \times 10^{-5}$). For prediction of South Asian ancestry, there were no significant differences among methods. For prediction of East Asian samples, BridgePRS was inferior to both PRS-CSx and PRS-CS-mult in both UKB and Bio*Me*, but these differences did not reach statistical significance.

Figure 4 shows the individual results for each trait ($R^2$ with confidence intervals) analyzed in the out-of-sample prediction into the Bio*Me* cohort. Although the methods showed similar results across many of the traits, the relative performance of the methods was highly variable, and for some traits there were distinct differences in the accuracy of the methods, especially in African ancestry samples. For example, in African ancestry samples, BridgePRS performed markedly better for mean corpuscular volume (MCV) and low-density lipoprotein (LDL), but markedly worse for eosinophil count. In both African and South Asian ancestry samples, the PRS-CSx prediction of height was highly inaccurate, possibly owing to the impact of variant nonoverlap between cohorts when applying PRS-CSx out of sample ('Discussion'). The corresponding trait-specific results for prediction into UKB are shown in Extended Data Fig. 5, with a similar pattern of results observed. Of note, BridgePRS again performed markedly better for MCV and LDL in African ancestry samples.

## Discussion

We have introduced a trans-ancestry PRS method, BridgePRS, that leverages shared genetic effects across ancestries to increase the accuracy of PRS in non-European populations. We benchmarked BridgePRS and the leading trans-ancestry PRS method PRS-CSx, as well as single-ancestry PRS methods PRS-CS and PRSice adapted for trans-ancestry prediction,

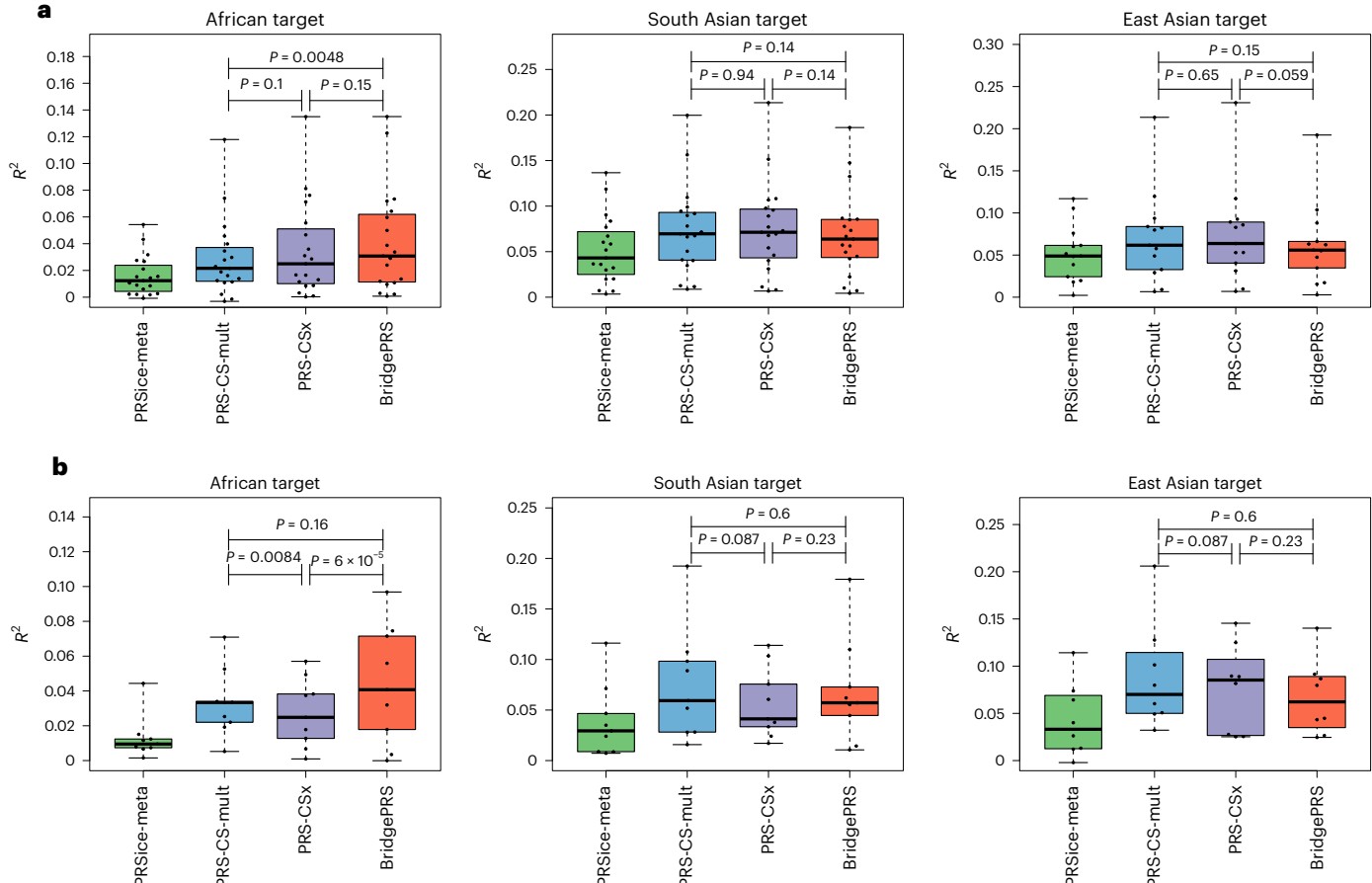

**Fig. 3 | Predictive accuracy of quantitative traits for different polygenic prediction methods and target populations. a,b,** Predictive accuracy, as measured by variance explained ($R^2$), of BridgePRS, PRS-CSx, PRS-CS-mult and PRSice-meta for African, South Asian and East Asian ancestry samples in UKB (**a**) and Bio*Me* (**b**). Nineteen and nine traits were included for African and South Asian ancestry samples in the UKB and Bio*Me* cohorts, respectively, and 13 and eight traits for East Asian samples in the UKB and Bio*Me* cohorts, respectively. The central rectangular boxes show the interquartile range, horizontal lines inside the boxes show the median, whiskers extend to the most extreme results and points show results for each trait. *P* values comparing methods were calculated as follows: for each trait, *z* statistics were calculated for the difference in $R^2$ between each pair of methods (the standard error of each $R^2$ estimate was estimated via bootstrapping using 10,000 replicates[20,21]). These *z* statistics were then summed to give Gaussian test statistics of mean 0 and variance (number of traits), under the null hypothesis of methods having the same $R^2$, from which two-tailed *P* values were derived.

across a range of simulated and real data. In all analyses, target population PRS used GWAS summary statistics from Europeans and the target population. Results from our simulated data suggest that BridgePRS has higher performance relative to PRS-CSx when the uncertainty is greater: for lower heritability traits, for lower GWAS sample sizes, when the genetic signal is dispersed over more causal variants (higher polygenicity), for greater between-population diversity (for example, with European base and African target rather than Asian target) and when the causal variants are not included in the analyses. In all analyses of simulated data, BridgePRS and PRS-CSx had superior performance relative to the single-ancestry PRS methods.

Application of the methods to real GWAS summary statistics from the UKB and BBJ cohorts and validation in independent samples of African, South Asian and East Asian ancestry in the UKB and Bio*Me* Biobank (recruited in the New York City area of the USA) gave results consistent with the simulations. Specifically, BridgePRS had superior average $R^2$ across the traits analyzed for samples of African ancestry, in which uncertainty was high owing to greater differences in LD between Africans and Europeans, and because of the relatively small African GWAS used. Likewise, PRS-CSx had superior average $R^2$ for samples of East Asian ancestry, for which differences in LD are smaller and the contributing East Asian GWASs are much larger (90,000–160,000). For

prediction into South Asian ancestry, in which LD is relatively similar but the South Asian GWASs used are small, the methods performed similarly.

The stronger performance of PRS-CSx in the real data analysis of East Asian samples may also have been due to PRS-CSx not requiring GWAS to be on the same scale and thus being unaffected by the rescaling of the BBJ effect estimates. PRS-CSx is unaffected by GWAS scale as it combines information across ancestries on the shrinkage (to zero) of the effect estimate of each SNP and does not combine information on the effect sizes. The final PRS-CSx PRS estimate is derived by combining ancestry-specific PRS with relative weights estimated in a linear regression in the test data. Differences in scale between the base GWAS are accounted for by the linear regression weights. BridgePRS should have improved performance when the GWASs used are performed on the same scale, as it shares information on effect sizes across ancestries.

In UKB and Bio*Me* data, we have demonstrated that BridgePRS has superior out-of-cohort predictive accuracy in genetic prediction in individuals of African ancestry. However, PRS-CSx has better accuracy when using UKB European and BBJ East Asian summary statistics to predict into individuals of East Asian ancestry. In general, in simulated and real data, BridgePRS performs better than PRS-CSx when uncertainty in mapping of causal variants is higher. Given the complementary

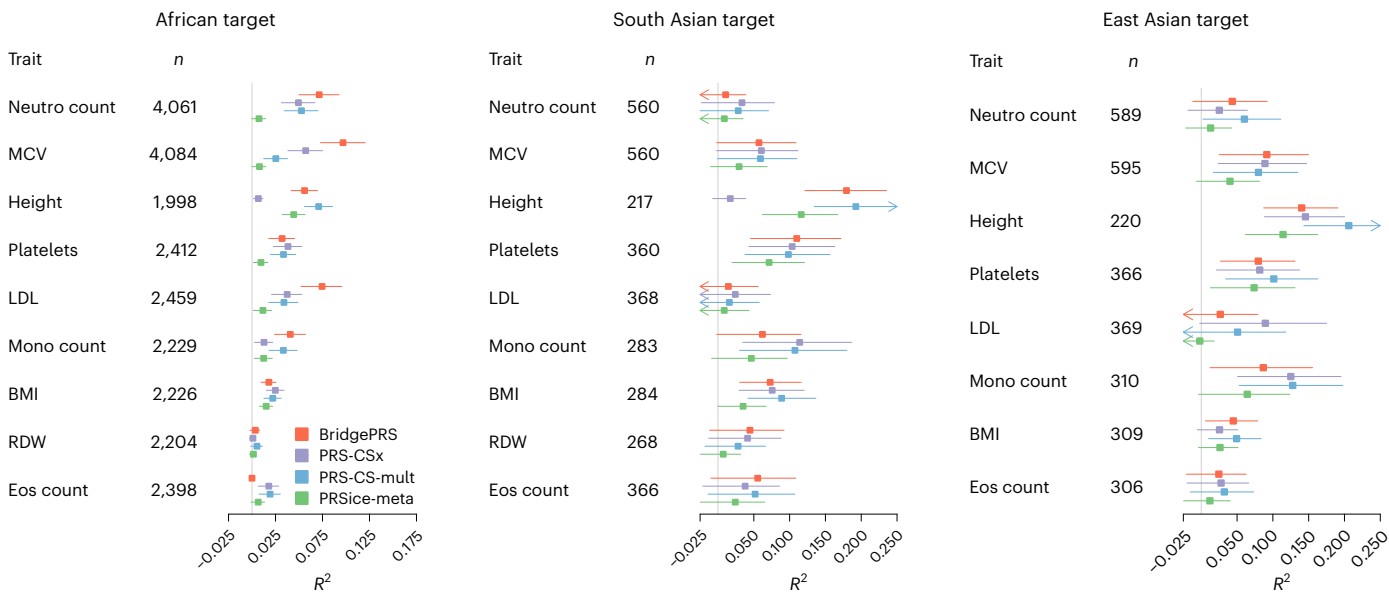

**Fig. 4 | Predictive accuracy of quantitative traits in Bio*Me* samples.** For each trait, variance explained ($R^2$), point estimates and 95% confidence intervals for BridgePRS, PRS-CSx, PRS-CS-mult and PRSice-meta are shown for African, South Asian and East Asian ancestry samples. Confidence intervals were calculated by bootstrapping using 10,000 replicates[20,21]. *n*, sample size; Neutro count, neutrophil count; MCV, mean corpuscular volume; Platelets, platelet count; Mono count, monocyte count; BMI, body mass index; RDW, red blood cell distribution width; Eos count, eosinophil count.

nature of the two methods, either can be optimal depending on the trait and study characteristics; therefore, we recommend applying both methods until it is known which offers greater power in the given setting.

BridgePRS is a fully dedicated PRS tool that performs the entire PRS process, is computationally efficient based on conjugate prior–posterior updates and offers a theoretical approach to tackling the PRS portability problem, with particularly strong performance for deriving PRS in populations of African and other underrepresented ancestries.

## Online content

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

## Methods

### The BridgePRS model

All modeling is performed at the locus-level, and each locus is assumed to be independent of all others. A locus is defined as a genomic region that captures all variants with $r^2 > 0.01$ within 1 Mb of a lead variant. Within loci, SNP effect sizes $\beta$ are modeled by a multivariate Gaussian distribution, and we assume that the trait $y$ of individuals with genotype data $X$ at the locus follows a Gaussian distribution $y \sim N(X\beta, \psi I)$. Throughout, the Gaussian distribution is parameterized by its mean and precision matrix (inverse covariance matrix).

Below, we describe the BridgePRS methodology used to derive a PRS for a target population, population 2 (in our applications: African, South Asian and East Asian) for which we have summary statistics from a relatively underpowered GWAS, and GWAS summary statistics from a well-powered GWAS from a different ancestral population, population 1 (in our applications: European). We also assume that we have small datasets of individual-level genotype–phenotype data from both populations.

**Stage 1: PRS informed by a single population.** In stage 1 modeling, we train and optimize PRS using GWAS summary statistics and test genotype–phenotype data from a single population. To determine the PRS for population 2, this modeling stage is applied to populations 1 and 2 (model 1 in Fig. 1). Application to population 1 determines the prior distributions for population 2 SNP effect sizes used in stage 2 (see below). Application of stage 1 modeling to population 2 only (model 2 in Fig. 1) is used to identify effects specific to population 2 that are missed when using population 1 effects as a prior.

In stage 1, a zero-centered conjugate Gaussian prior is assigned for the SNP effects at each locus $\beta \sim N(0, \psi(\text{diag}(\lambda)))$, where $\lambda$ is a vector of SNP-specific shrinkage parameters. The use of a conjugate prior allows the posterior distribution of SNP effects to be determined analytically[22]:

$$\beta \sim N\left((\text{diag}(\lambda) + X^T X)^{-1} X^T y, \psi(\text{diag}(\lambda) + X^T X)\right).$$

$X^T y$ can be calculated from the vector of maximum likelihood marginal effects, $\hat{\beta}$, available from GWAS summary statistics by $(X^T y)_i = 2n\theta_i(1 - \theta_i)\hat{\beta}_i$, where $n$ is the sample size, $\theta$ is the vector of allele frequencies and $(X^T y)_i$ is the $i$th element of $X^T y$, with $i$ indexing SNPs. $X^T X = n\Phi$; here, $\Phi$ is the pairwise genotypic covariance, which can be estimated from a reference panel representative of the population used in the GWAS. Thus, rescaling $\lambda$ by $n$, the posterior is estimated as

$$\beta \sim N\left((\text{diag}(\lambda) + \Phi)^{-1}\theta(1 - \theta)\hat{\beta}, \psi(\text{diag}(\lambda) + \Phi)\right)$$

$$\beta \sim N\left(\tilde{\beta}, \psi\Omega\right).$$

To accommodate the effects of natural selection, we allow the prior on SNP effects to be dependent on allele frequencies such that the prior precision for the $k$th SNP is $\lambda^{(k)} = \lambda^{(0)}(\theta_k(1 - \theta_k))^{\alpha}$ and $\alpha \in [0, 1]$ (ref. 23). When $\alpha = 0$, allele frequencies and effect size are a priori independent. $\alpha = 1$ is the value implicitly assumed by many methods[24] and implies a strong assumption of larger effects at SNPs of lower minor allele frequency. Multiple models are fit at each locus under priors defined by all combinations of $\alpha = (0, 0.25, 0.5, 0.75, 1)$ and $\lambda^{(0)} = (0.05, 0.1, 0.2, 0.5, 1, 2, 5)$. Loci are ranked by the $P$ value of their most associated SNP and assigned to subset $S_k$; if the top SNP $P$ value is less than $10^{-k}$, values of $k = 1, \ldots, 8$ are considered. Multiple genome-wide PRSs are calculated for a test set of phenotype and genotype data by summing the effects across all contributing loci for all combinations of $\alpha$, $\lambda_0$ and $k$:

$$\text{PRS}_{ijk} = \sum_{l \in S_k} X_l \tilde{\beta}^{(l)}_{\lambda^{(0)}_i \alpha_j},$$

where $X_l$ is the genotype data at locus $l$, $\hat{\beta}^{(l)}_{\lambda^{(0)}_i \alpha_j}$ is the posterior mean at locus $l$ with prior defined by parameters $\lambda^{(0)}_i$ and $\alpha_j$, and $S_k$ is the subset

of loci with top SNP $P$ value $<10^{-k}$. A single PRS is calculated by a weighted sum of the PRS across all $i, j$ and $k$, with weights determined by a ridge regression fit to the test data, using leave-one-out cross-validation to select the ridge shrinkage parameter that minimizes out-of-sample deviance, as implemented in the R package glmnet[25].

**Stage 2: PRS informed by stage 1.** In stage 2 modeling, SNP effect sizes estimated by the application of stage 1 modeling to population 1 (for example, Europeans) are updated based on population 2 GWAS summary statistics and optimized using population 2 genotype–phenotype data. The prior used is taken as the posterior derived from the $\lambda_0$ and $\alpha$ prior parameters, which optimize prediction in the test data of population 1. As for stage 1, this prior is also a multivariate Gaussian. A parameter $\tau$ is added to the precision parameter of the Gaussian to control the contribution of population 1 to population 2; thus, the prior is specified as $\beta_2 \sim N(\tilde{\beta}_1, \psi\tau\Omega_1)$. This is similarly a conjugate model with a Gaussian posterior[22]:

$$\beta_2 \sim N\left((\tau\Omega_1 + \Phi_2)^{-1}(\tau\Omega_1\tilde{\beta}_1 + \hat{\beta}_2\theta_2(1 - \theta_2)), \psi(\tau\Omega_1 + \Phi_2)\right)$$

$$\beta_2 \sim N(\tilde{\beta}_2, \Omega_2),$$

where $\Phi_2$ is the SNP covariance at the locus in population 2, $\hat{\beta}_2$ is the vector of marginal maximum likelihood SNP effect sizes and $\theta_2$ is the vector of allele frequencies. Small values of $\tau$ correspond to using effect estimates close to those from population 2. As $\tau$ increases, more weight is assigned to population 1, such that as $\tau \to \infty$, $\beta_2 \to \beta_1$.

**Ranking loci in stage 2.** Owing to differences in LD between populations, we do not rank loci by the $P$ value of a single best SNP but instead aggregate information across loci by adapting the $F$ test. We show below that the $F$ test in a multivariate linear regression model for the null $H_0$: $\beta = 0$ is well approximated by:

$$F_{\text{stat}} = \frac{n - k}{kn\sigma^2} \beta^T X^T X \beta$$

with degrees of freedom $k$ and $n - k$, where $k$ is the dimension of $\beta$, $n$ is the number of observations and $\sigma^2$ is the phenotypic variance. The maximum likelihood estimate and $X^T X$ are substituted by the posterior mean and precision matrix and $n$ with $n_{\text{eff}} = n(1 + \tau)$, the effective number of observations accounting for the prior, giving the statistic:

$$F_{\text{Bayes}} = \frac{n_{\text{eff}} - k}{k\sigma^2} \tilde{\beta}_2 \Omega_2 \tilde{\beta}_2.$$

The resultant tail probability is analogous to a $P$ value, although it cannot be interpreted as such as the parameter estimates $\beta$ and $\lambda$ include prior information. Instead, for each $\tau$, a locus with test statistic $F$ is assigned to $S_k$ if $F > q_k$, where $q_k$ is the $F$ quantile corresponding to Prob($p < 10^{-k}$), where the values $p$ are the locus-specific top SNP $P$ values. This ranking ensures that the pseudo $F$ statistic ranking assigns the same number of loci to each subset as the SNP $P$ value ranking. As for the stage 1 single-ancestry PRS, multiple genome-wide PRSs are constructed by:

$$\text{PRS}_{ik} = \sum_{l \in S_k} X_l \tilde{\beta}^{(l)}_{\tau_i},$$

where $\tilde{\beta}^{(l)}_{\tau_i}$ is the posterior mean at locus $l$ with prior defined by parameter $\tau_i$, and $S_k$ is the subset of loci with $F > q_k$. Models are fit for $\tau = 1, 2, 5, 10, 15, 20, 50, 100, 200$ and $500$ and the same $P$ value thresholds as those used in stage 1 of the modeling. A single PRS is estimated via a ridge regression fit using population 2 test data as described above using glmnet.

Supplementary Table 1 shows the average $R^2$ from BridgePRS ranking loci by the pseudo $F$ statistic versus the $P$ value from the European GWAS across the 19 traits analyzed here for African and South Asian

UKB samples. There were broadly similar results for the pseudo $F$ statistic versus the $P$ value ranking: 0.0413 versus 0.0403 and 0.0683 versus 0.0688 in African and South Asian samples, respectively. Also shown in Supplementary Table 1 are equivalent results using UKB genotyped variants (rather than imputed variants); here, there was a pronounced improvement using the pseudo $F$ statistic ranking: 0.0413 versus 0.0359 in African samples and 0.0694 versus 0.0646 in South Asian samples ($P = 0.086$ for the superiority of the $F$ statistic ranking). All results presented here were obtained using the pseudo $F$ statistic loci ranking. The BridgePRS software allows users to rank loci in stage 2 using either of the two ranking methods.

**Incomplete SNP overlap between populations 1 and 2.** Quality control (QC) is performed separately in each population; see below. This results in variants included in analyses differing between populations. Thus, stage 2 analyses are performed on the intersection of variants passing QC in both populations and the prior is calculated conditional on effects of nonoverlapping variants set to zero. Thus, given a prior of $\beta_2 \sim N(\bar{\beta}_1, \psi\tau\Omega_1)$, the prior on the overlapping variants is given by[22]

$$p\left(\beta_2^{(a)}|\beta_2^{(b)} = 0\right) = N\left(\bar{\beta}_1^{(a)} + \left(\Omega_1^{(aa)}\right)^{-1}\Omega_1^{(ab)}\bar{\beta}_1^{(b)}, \psi\tau\Omega_1^{(aa)}\right),$$

where $a$ represents the overlapping variants and $b$ the nonoverlapping variants, and $\Omega_1^{(aa)}$ and $\Omega_1^{(ab)}$ are the appropriate submatrices of $\Omega_1$. SNP overlap is taken at stage 2 to allow models fit in stage 1 to be applied to other datasets with different SNP sets.

**Combining PRSs.** We consider three alternative models for the PRS of population 2: (1) PRS estimated using only the two-stage European-informed PRS, that is, where the population 2 GWAS is underpowered and contributes insufficient information on its own; (2) PRS estimated using only population 2, that is, where European GWAS does not inform the PRS of population 2; and (3) the case where both the population-2-only PRS and the two-stage PRS contribute independent information. The estimation of models (1) and (2) is determined by a cross-validated ridge regression fit as described above using glmnet. Model (3) is estimated similarly by merging all single-ancestry and two-stage PRS and weighting by a cross-validated ridge regression fit.

The final PRS is a weighted sum of these three PRS, with weights determined by the estimated marginal likelihood of each. The log-marginal likelihood of a linear regression model $M_i$ can be approximated by[26]

$$\log p(y, X|M_i) = \frac{n}{2}\log\sigma_i^2 + \kappa,$$

where $\sigma_i^2$ is the residual model variance estimated from cross-validation and $\kappa$ is a constant. With equal prior weight for each of the models, the posterior model weights for models $M_1$, $M_2$ and $M_3$ are given by:

$$p(M_i|y, X) = \frac{\exp\{n\log\sigma_i^2/2\}}{\sum_{i=1}^{3}\exp\{n\log\sigma_i^2/2\}}.$$

Combining PRSs in this way can be extended to any number of contributing PRS. For example, we also combined PRSs for African ancestry samples constructed from East Asian BBJ and African UKB GWAS summary statistics to PRS constructed in our main analysis that used African and European UKB GWAS summary statistics. Supplementary Fig. 1 compares trait $R^2$ for African + European PRS with African + European + East Asian PRS for UKB and BBJ overlapping traits. Marginal improvement was observed with the addition of the BBJ East Asian data for monocyte count, BMI and height; for the other traits, $R^2$ was practically unaltered.

**Definition of loci.** Loci for the two-stage modeling were defined by clumping and thresholding of European GWAS summary statistics and LD estimated from UKB European samples using PLINK v.1.9 (ref. 27) with the following parameters: --clump-p1 0.01, --clump-p2 0.01, --clump-kb 1,000, --clump-r2 0.01. The $P$ value for each locus was determined by the $P$ value of the lead SNP of the locus in the European GWAS. The ancestry-specific loci were defined similarly but used GWAS and LD data from the appropriate ancestry.

**Estimating LD.** BridgePRS calculates LD on the fly using genotype data supplied by the user and is therefore not restricted to any predefined subset of variants. In the simulation analyses, BridgePRS used all 1,000G samples from the appropriate ancestry to estimate LD, and in the real data analyses a subsample (between 5,000 and 6,000) of UKB samples from the appropriate ancestry was used.

**Application of PRS-CSx**
PRS-CSx is a Python-based software package that integrates GWAS summary statistics and LD reference data from multiple populations to estimate population-specific PRS. PRS-CSx applies a continuous shrinkage prior to SNP effects genome-wide in which the sparseness of the genetic architecture across populations is controlled by a parameter $\phi$. PRS-CSx does not make any inference on $\phi$ but instead estimates separate PRS for each value of $\phi$ considered. Throughout, we followed the implementation described in Ruan et al.[5]; thus, values of $\phi = (10^{-6}, 10^{-4}, 10^{-2}$ and 1) were considered. For each $\phi$, PRS-CSx first estimates population-specific PRS, for example. $PRS_{\phi,EUR}$ (European) and $PRS_{\phi,AFR}$ (African), where $PRS_{\phi,x}$ is the standardized PRS for population $x$. For each $\phi$, PRS-CSx fits the following linear regression to the target population test data $y$:

$$y = w_{\phi,EUR}PRS_{\phi,EUR} + w_{\phi,AFR}PRS_{\phi,AFR} + e.$$

where $e$ is Gaussian error. The $\phi$ value and the corresponding regression coefficients for the linear combination of PRSs that maximize the coefficient of determination ($R^2$) in the target population (for example, African) test set were used in the validation dataset to calculate the final PRS:

$$PRS_{final} = \hat{w}_{\hat{\phi},EUR}PRS_{\hat{\phi},EUR} + \hat{w}_{\hat{\phi},AFR}PRS_{\hat{\phi},AFR}$$

Unlike BridgePRS, PRS-CSx does not use European test data to estimate non-European PRS. Therefore, to ensure that both methods used the same data, GWASs were performed on the European test samples using PLINK v.2.0 (ref. 27) and then meta-analyzed with the GWAS data from the European data METAL[28]. The meta-analyzed European GWAS, the GWASs generated from the training samples of the target population and the LD reference panel generated by the authors of PRS-CSx were provided to PRS-CSx.

**UKB genotype and sample QC**
The UKB is a prospective cohort study of around 500,000 individuals recruited across the United Kingdom during 2006–2010. The genetic data comprise 488,377 samples genotyped at 805,426 SNPs. Population ancestries were defined by four-means clustering performed on the first two principal components (PCs) of the genotype data. The ancestry of each cluster was defined by the country of birth (field ID: 20115) of the majority of individuals in the cluster. Standard QC procedures were then performed on each ancestry cluster independently; any SNP with minor allele frequency <0.01, genotype missingness >0.02 or Hardy–Weinberg equilibrium test $P$ value < $10^{-8}$ was removed. Samples with high levels of missingness or heterozygosity, with mismatching genetic-inferred and self-reported sex, or with aneuploidy of the sex chromosomes were removed as recommended by the UKB data processing team. A greedy algorithm[29] was used to remove related

individuals, with kinship coefficient >0.044, in a way that maximized sample retention. In total, 557,369 SNPs and 387,392 individuals were retained for analysis.

## Imputation

Imputed variants were extracted from imputed UKB data using PLINK v.2.0, converting the imputed data into hard-coded genotypes and retaining variants with the following filters: biallelic variants (--max-alleles 2), minor allele frequency greater than 0.001 (-maf 0.001), genotype missingness less than 1% (--geno 0.01) and MACH info score greater than 0.8 (--mach-r2-filter 0.8).

## Trait selection

We extracted all continuous traits from unique samples in the UKB and performed basic filtering, discarding samples with phenotypic values six standard deviations away from the mean. Traits with more than 2,000 samples of African ancestry were extracted. For each trait, 300,000 European samples were extracted (retaining at least 10,000 samples for test and validation for each trait) and GWASs were run on the genotype data using PLINK v.2.0 with --glm. Sex (field ID: 31), age (field ID: 21003), genotyping batch, UKB assessment center (field ID: 54) and 40 PCs were included as covariates, with fasting time (field ID: 74) and dilution factor (field ID: 30897) also included for blood biochemical traits. LD score regression[30] was run on the resultant summary statistics and traits were further filtered, discarding those with heritability less than 1%. The remaining traits were ranked according to their heritability, and traits correlated with a more heritable trait (absolute Pearson correlation greater than 0.3) were removed, resulting in 27 traits. Results are presented for 19 traits that had an $R^2$ in Africans of greater than 1% for at least one analysis. The sample sizes for each trait and ancestry are shown in Extended Data Table 1.

## Implementation

European, African and South Asian UKB samples were split into three independent groups: training data to construct the GWAS summary statistics, test data to select best-fitting parameters, and validation data to calculate out-of-sample predictive accuracy. The proportions of samples allocated to each set were 2/3 training, 1/6 test and 1/6 validation. Each GWAS was run in PLINK v.2.0 as described above. East Asian samples were split equally between test and validation sets.

For each trait, analyses were run with imputed variants. GWASs were run separately for the training samples of European, African and South Asian ancestry for each of the 19 traits using PLINK v.2.0 as described above. All PRSs were calculated using two populations: the African PRS used African and European UKB GWAS data, the South Asian PRS used South Asian and European UKB GWAS data, and the East Asian PRS used BBJ and European UKB GWAS.

## Application to BioMe

BioMe samples were genotyped on the Infinium Global Screening Array v.1.0 platform. Samples were removed if they had a population-specific heterozygosity rate of greater than ±6 standard deviations of the population-specific mean, along with a call rate of <95%. In addition, samples were removed if they exhibited persistent discordance between the electronic health record and genetic sex. Variants were removed that had a call rate <95%, a Hardy–Weinberg Equilibrium $P$ value threshold of $P < 10^{-5}$ in African American and European American ancestry, or $P < 10^{-13}$ in Hispanic and South Asian ancestry.

PC analysis was performed, and African, South Asian and East Asian samples were selected by clusters on PC plots corresponding to self-reported ancestry. African samples were selected as those with PC1 > 0.0075, PC2 < −0.0005 and PC3 > −0.002. South Asian samples were selected as those with −0.01 < PC3 < −0.004, −0.003 < PC4 < 0.001 and PC5 < −0.015. East Asian samples were selected as those with PC3 < −0.01, PC4 > 0.001, PC5 > −0.005 and PC6 > −0.0035.

Supplementary Figs. 2–4 plot the top six PCs, with samples colored by self-reported ancestry, and show the thresholds used to select African, South Asian and East Asian ancestry samples.

Imputation was performed using IMPUTE2 (ref. 31) with the 1000G Phase 3 v.5 reference panel[12]. Variants were first filtered by info score >0.3. Genotype data for the calculation of PRS in unique individuals were generated for in each of the two ancestry groups separately by first removing variants with minor allele frequency <1% in the respective BioMe population and then removing one of each pair of variants with duplicate genomic position. BioMe variants were mapped onto the UKB PRS by genomic position (build 37). Variants were coded by their expected allele count (dosage) for the calculation of PRS. Samples with phenotypic values three standard deviations away from the mean were excluded.

## Measure of PRS accuracy

Variance explained was calculated as

$$R^2 = 1 - \frac{\text{Var}_{(y|M_1)}}{\text{Var}_{(y|M_0)}},$$

where $M_i$ is the regression model with ($i = 1$) and without ($i = 0$) the PRS, with both models including covariates for the top 40 PCs, age, sex, center and batch, fasting and dilution for the biochemical traits. Variance explained in the applications to BioMe included covariates for age, sex and the top 32 PCs. Standard errors and confidence intervals were calculated by bootstrapping in the R package boot[20,21] using 10,000 replicates.

## Equivalence of sample size and heritability on GWAS power

We assume a phenotype value is given by additive genetic effects $\beta$ and an environmental component $e$

$$Y = \sum_j X_j \beta_j + e,$$

where $e \approx N(0, \sigma_e^2)$. Therefore,

$$\text{Var}(Y) = \sum_j \beta_j^2 \text{Var}(X_j) + \sigma_e^2;$$

setting variance due to genetics to $\sigma_g^2$, we have

$$= \sigma_g^2 + \sigma_e^2.$$

As heritability $h^2 = \frac{\sigma_g^2}{\text{Var}(Y)}$, for fixed genetic effects $\beta$ and therefore fixed $\sigma_g^2$, if heritability changes by a factor of $\kappa$, $\text{Var}(Y)$ must change by a factor of $\kappa^{-1}$. If the genetic effect $\beta_j$ in a GWAS is estimated in a linear regression model, the expected variance of its maximum likelihood estimate $\hat{\beta}_j$ is approximately $\frac{\text{Var}(Y)}{n\text{Var}(X)}$. Therefore, changing $h^2$ by a factor of $\kappa$, and thus $\text{Var}(Y)$ by a factor of $\kappa^{-1}$, has the same effect on $\text{Var}(\hat{\beta}_j)$ as changing the sample size $n$ by a factor of $\kappa$.

## Reformulation of the F test

Without loss of generality, assume zero-centered normally distributed trait data $y$ with variance $\sigma^2$. A linear regression is fitted to this data with an $n \times k$ covariate matrix $X$, resulting in maximum likelihood estimates $\hat{\beta}$. The $F$ statistic is defined by the residual sum of squares of the null and alternative models ($\text{RSS}_0$ and $\text{RSS}_1$) as follows:

$$F = \frac{n-k}{k} \left( \frac{\text{RSS}_0 - \text{RSS}_1}{\text{RSS}_1} \right)$$

$$= \frac{n-k}{k} \left( \frac{y^T y - (y - X\hat{\beta})^T (y - X\hat{\beta})}{(y - X\hat{\beta})^T (y - X\hat{\beta})} \right)$$

$$= \frac{n-k}{k} \left( \frac{n\sigma^2}{(y - X\hat{\beta})^T (y - X\hat{\beta})} - 1 \right)$$

$$= \frac{n-k}{k} \left( \frac{n\sigma^2}{y^T y - \hat{\beta}^T X^T y - y^T X\hat{\beta} + \hat{\beta}^T X^T X\hat{\beta}} - 1 \right),$$

as $\hat{\beta} = (X^T X)^{-1} X^T y$

$$= \frac{n-k}{k} \left( \frac{n\sigma^2}{n\sigma^2 - \hat{\beta}^T X^T X \hat{\beta}} - 1 \right)$$

$$= \frac{n-k}{k} \left( \frac{\hat{\beta}^T X^T X \hat{\beta}}{n\sigma^2 - \hat{\beta}^T X^T X \hat{\beta}} \right)$$

$$= \frac{n-k}{kn\sigma^2} \hat{\beta}^T X^T X \hat{\beta} \left( 1 - \frac{1}{n\sigma^2} \hat{\beta}^T X^T X \hat{\beta} \right)^{-1}.$$

$\frac{\hat{\beta}^T X^T X \hat{\beta}}{\sigma^2}$ is the variance explained by the locus; therefore, assuming this is small, a first-order Taylor approximation can be used to give

$$\approx \frac{n-k}{kn\sigma^2} \hat{\beta}^T X^T X \hat{\beta}.$$

### Reporting summary
Further information on research design is available in the Nature Portfolio Reporting Summary linked to this article.

### Data availability
Publicly available data used to generate the simulated data are available from the following sites. 1000G Phase 3 reference panels: https://mathgen.stats.ox.ac.uk/impute/1000GP_Phase3.html; and genetic maps for each subpopulation: ftp.1000genomes.ebi.ac.uk/vol1/ftp/technical/working/20130507_omni_recombination_rates. UKB genotype and phenotype data were obtained from the UKB resource under application 18177 (https://www.ukbiobank.ac.uk/enable-your-research/approved-research/multi-trait-gwas-analyses-in-the-uk-biobank). UKB QC information (missingness, allele frequency, Hardy–Weinberg equilibrium) was obtained from UKB resource 531 (https://biobank.ctsu.ox.ac.uk/crystal/refer.cgi?id=531). Recruitment and enrollment of participants into Bio*Me* was Institutional Review Board (IRB) and Health Insurance Portability and Accountability Act 1996 (HIPAA) approved. It is a biobank linked to electronic medical records that allows the use of deidentified samples linkable to past, present and future clinical information from electronic health records at Mount Sinai. Bio*Me* contains protected health information and is thus under controlled access. Applications to access the data can be made to biome@mountsinai.org; see also https://icahn.mssm.edu/research/ipm/programs/biome-biobank. BBJ summary statistics were downloaded from PheWeb: https://pheweb.jp. SNP weights for the polygenic risk scores estimated by BridgePRS in this paper are available on GitHub (https://github.com/clivehoggart/BridgePRS_data).

### Code availability
Software, example data and a tutorial for BridgePRS are available from www.bridgeprs.net. Source code, to which www.bridgeprs.net links, is available from https://github.com/clivehoggart/BridgePRS, DOI badge https://doi.org/10.5281/zenodo.8385983, v.0.1 (ref. 32). Scripts used for all analyses are available on GitHub: https://github.com/clivehoggart/BridgePRS_data. All other code used in this study is available from the following websites: BridgePRS: https://www.bridgeprs.net; HAPGEN2 v.2.2.0: https://mathgen.stats.ox.ac.uk/genetics_software/hapgen/hapgen2.html; IMPUTE2 v.2: https://mathgen.stats.ox.ac.uk/impute/impute_v2.html; LDSC v.1.0.1: https://github.com/bulik/ldsc; METAL v.2011-03-25: http://csg.sph.umich.edu/abecasis/metal/; PLINK v.1.9: https://www.cog-genomics.org/plink; PLINK v.2.0: https://www.cog-genomics.org/plink/2.0/; PRS-CSx v.1.0.0: https://github.com/getian107/PRScsx; PRS-CS v.1.0.0: https://github.com/getian107/PRScs; PRSice-2 v.2: https://www.prsice.info; R v.4.0.3: https://cran.r-project.org; R boot package v.1.3.25: https://cran.r-project.org/web/packages/boot/index.html; Ridge reg glmnet package v.4.0-2: https://cran.r-project.org/web/packages/glmnet/index.html.

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

### Acknowledgements
We thank the participants in the UK Biobank (UKB), Biobank Japan (BBJ) and Bio*Me* Biobank and the scientists involved in the construction of these resources. This research has been conducted using the UKB resource under application 18177 (P.F.O.). All participants gave full informed consent. This work was supported by grants to P.F.O. from the National Institute of Mental Health (R01MH122866) and the National Human Genome Research Institute (R01HG012773) and through the computational resources and staff expertise provided by Scientific Computing at the Icahn School of Medicine at Mount Sinai, in particular, the Minerva and Data Ark teams. We also thank A. Ori, B. Rowan, C. Iyegbe, H. M. (Beatrice) Wu, L. Liou, L. Sloofman and Z. Wang for helpful discussions.

### Author contributions
C.J.H. and P.F.O. conceived and designed the project and methodology. C.J.H. developed the statistical modeling with input from P.F.O. C.J.H. programmed all the BridgePRS code and performed the analyses. S.W.C. preprocessed the UKB data and performed the GWAS in the UKB. S.W.C. and J.G.-G. developed the pipeline to run PRS-CSx on the data. T.S. tested the code and wrote a wrapper for the software. T.S. also developed the BridgePRS software website, with input from C.J.H. and P.F.O. M.P. preprocessed the Bio*Me* data. C.J.H. and P.F.O. wrote the manuscript, and all authors reviewed and approved the final version.

### Competing interests
S.W.C. is a current employee of Regeneron Genetics Center. The other authors declare no competing interests.

### Additional information
**Extended data** is available for this paper at https://doi.org/10.1038/s41588-023-01583-9.

**Correspondence and requests for materials** should be addressed to Clive J. Hoggart or Paul F. O'Reilly.

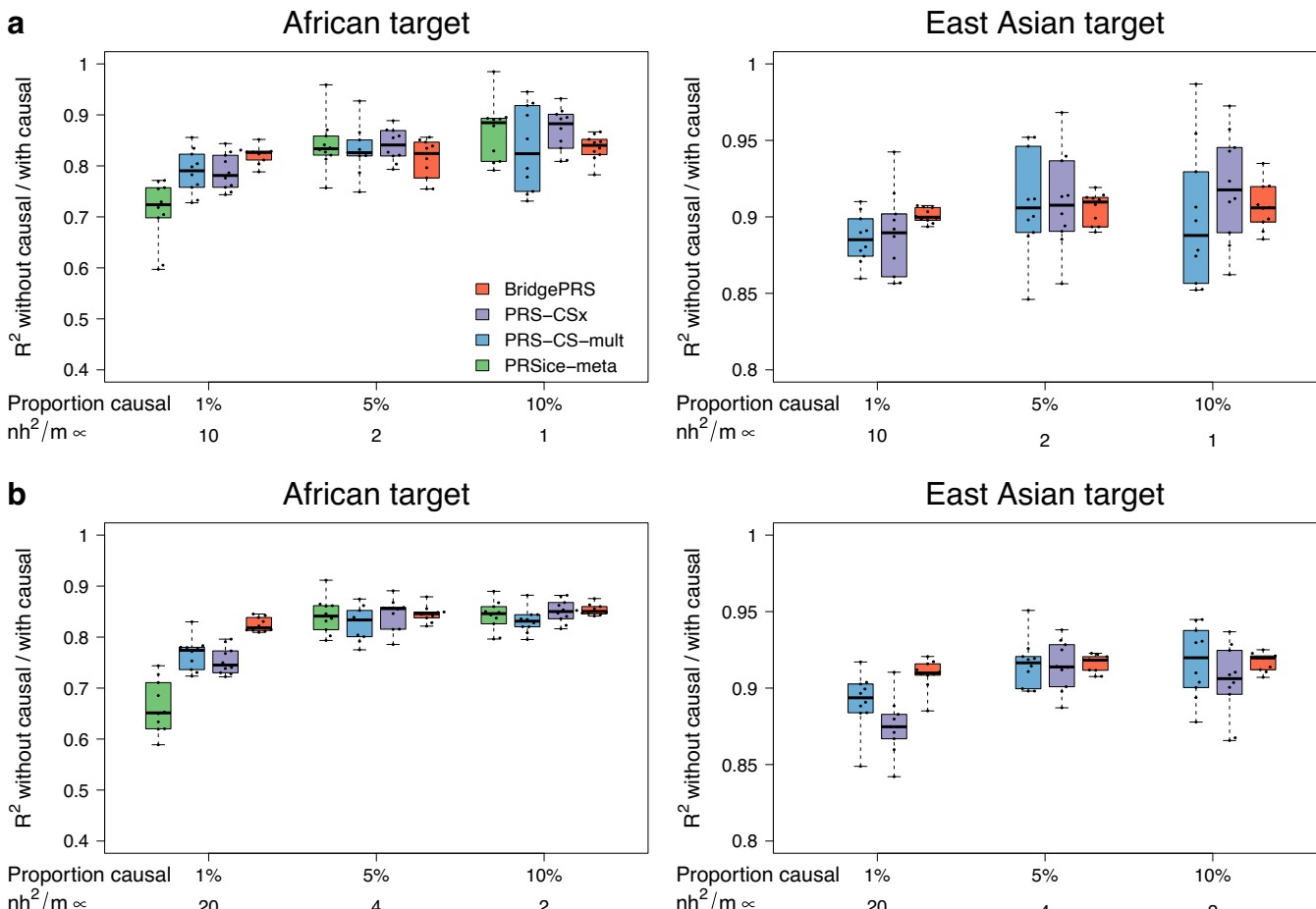

**Extended Data Fig. 1 | Relative loss in removing causal variants from analysis in simulated data.** Relative loss measured by ratio of models' variance explained ($R^2$) without and with the causal variants included. Results are shown for BridgePRS, PRS-CSx, PRS-CS-mult and PRSice-meta across six simulation scenarios for African and East Asian ancestry samples. **a** SNP heritability $h^2_{SNP} = 0.25$ and **b** SNP heritability $h^2_{SNP} = 0.5$, ten simulated phenotypes per scenario. Under each set of analyses the proportion of causal variants and the relative power of the data used is shown, measured by $nh^2/m$ up to proportionality, where n is the GWAS sample size, $h^2$ heritability and m the number of causal variants. The central rectangular boxes show the interquartile range, horizontal lines inside the boxes show the median, whiskers extend to the most extreme results and points show results for each of the 10 simulated phenotypes. PRSice-meta results for East Asian analyses were unstable and removed for clarity.

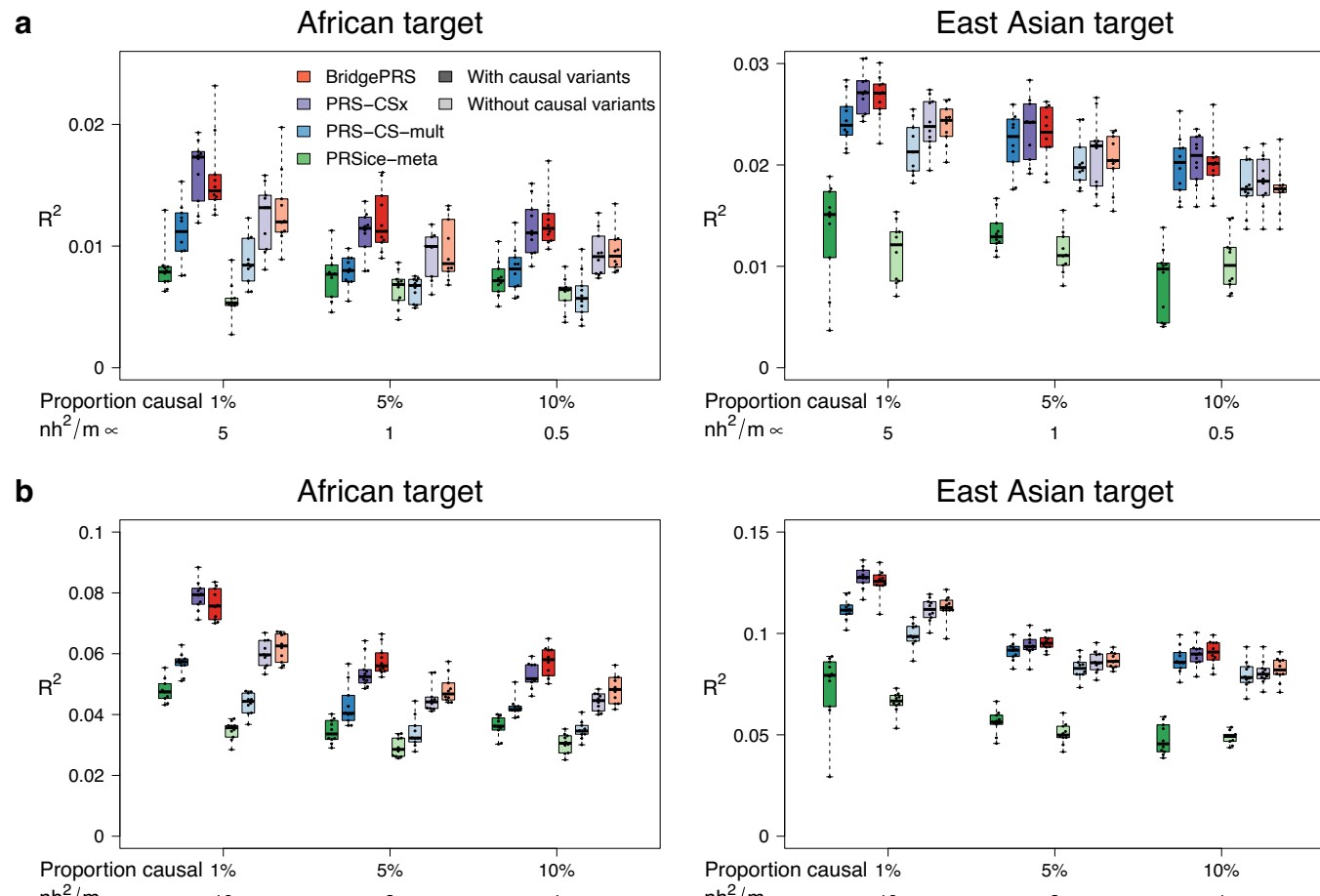

**Extended Data Fig. 2 | Predictive accuracy for different polygenic prediction methods in simulations using half GWAS sample size as used in the primary simulation.** Sample sizes of 40K European and 10K non-European were used. Results are shown for BridgePRS, PRS-CSx, PRS-CS-mult and PRSice-meta across six simulation scenarios, with and without the causal variants included in the model for African and East Asian ancestry samples. **a** SNP heritability $h^2_{SNP} = 0.25$ and **b** SNP heritability $h^2_{SNP} = 0.5$, ten simulated phenotypes per scenario. Under each set of analyses the proportion of causal variants and the relative power of the data used is shown, measured by $nh^2/m$ up to proportionality, where n is the GWAS sample size, $h^2$ heritability and m the number of causal variants. The central rectangular boxes show the interquartile range, horizontal lines inside the boxes show the median, whiskers extend to the most extreme results and points show results for each of the 10 simulated phenotypes.

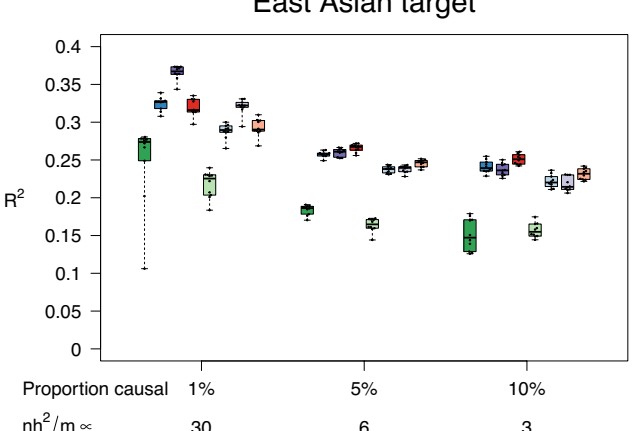

**Extended Data Fig. 3 | Predictive accuracy for different polygenic prediction methods in simulations at $h^2_{SNP} = 0.75$.** Results are shown for BridgePRS, PRS-CSx, PRS-CS-mult and PRSice-meta across six simulation scenarios, with and without the causal variants included in the model for African and East Asian ancestry samples, ten simulated phenotypes per scenario. Under each set of analyses the proportion of causal variants and the relative power of the data used is shown, measured by $nh^2/m$ up to proportionality, where n is the GWAS sample size, $h^2$ heritability and m the number of causal variants. The central rectangular boxes show the interquartile range, horizontal lines inside the boxes show the median, whiskers extend to the most extreme results and points show results for each of the 10 simulated phenotypes.

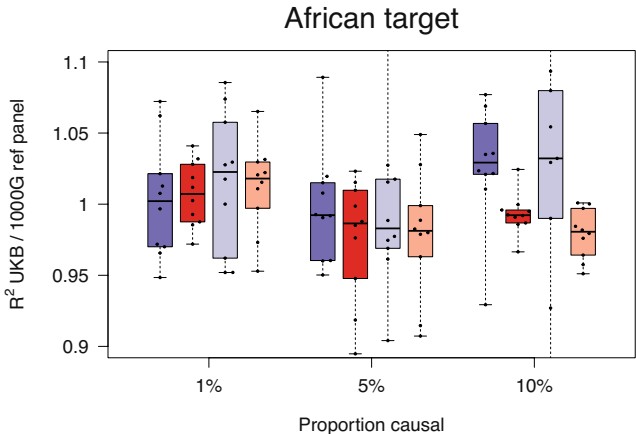

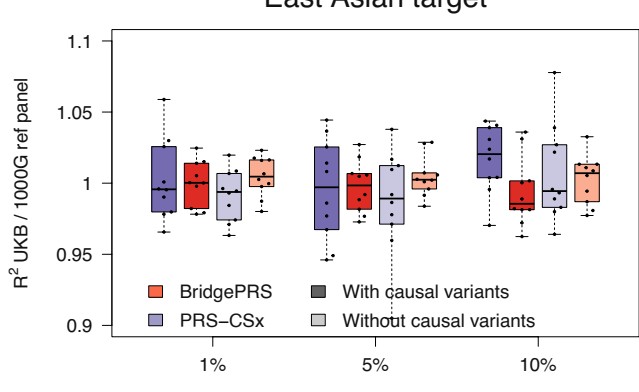

**Extended Data Fig. 4 | Ratio of phenotypic variance explained $R^2$ using UK Biobank and 1000 Genomes LD reference panels in simulations.** Results are shown for BridgePRS and PRS-CSx across six simulation scenarios, 10 simulated phenotypes per scenario with $h^2$=0.25 for African and East Asian ancestry samples. Data was simulated using 1000 Genomes as reference. Under each set of analyses the proportion of causal variants and the relative power of the data used is shown, measured by $nh^2/m$ up to proportionality, where n is the GWAS sample size, $h^2$ heritability and m the number of causal variants. The central rectangular boxes show the interquartile range, horizontal lines inside the boxes show the median, whiskers extend to the most extreme results and points show results for each of the 10 simulated phenotypes.

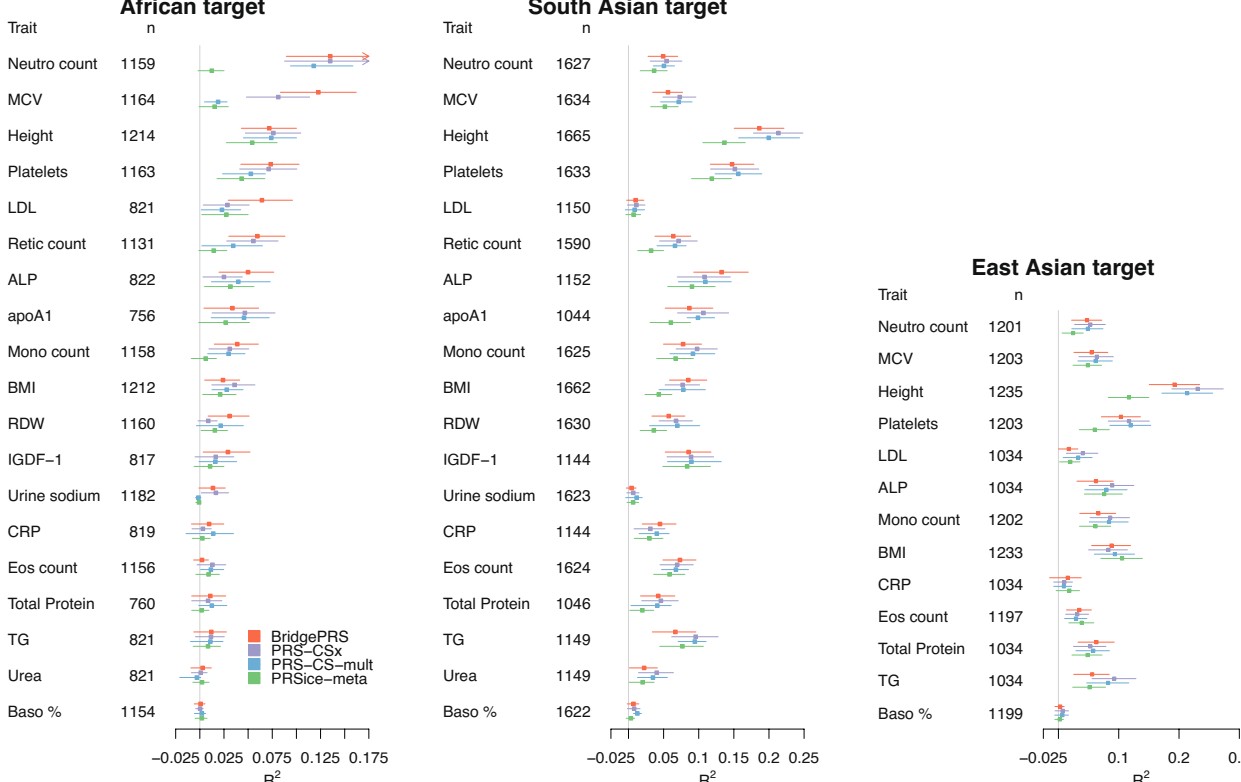

**Extended Data Fig. 5 | Predictive accuracy for quantitative traits in UK Biobank samples.** For each trait variance explained ($R^2$), point estimates and 95% confidence intervals, by BridgePRS, PRS-CSx, PRS-CS-mult and PRSice-meta are shown for African, South Asian and East Asian ancestry samples. n indicates sample size. Neutro count=Neutrophil count, MCV=Mean corpuscular volume, Platelets=Platelet count, Retic count=Reticulocyte per- centage, ALP=Alkaline phosphatase, Mono count=Monocyte count, apoA1=Apolipoprotein A, BMI=Body mass index, RDW=Red blood cell distribution width, Eos count=Eosinophil count, TG=Triglycerides, Baso %=Basophil percentage, CRP=C-reactive protein. Confidence intervals were calculated by bootstrapping using 10,000 replicates.

## Extended Data Table 1 | GWAS and UK Biobank test and validation sample sizes

| | European | African | | | South Asian | | | East Asian | | |
|---|---|---|---|---|---|---|---|---|---|---|
| | Train | Train | Test | Valid. | Train | Test | Valid. | Train* | Test | Valid. |
| Height | 257675 | 4862 | 1215 | 1214 | 6666 | 1666 | 1665 | 191787 | 1235 | 1235 |
| BMI | 257327 | 4853 | 1213 | 1212 | 6653 | 1662 | 1662 | 173430 | 1234 | 1233 |
| Platelet # | 250404 | 4657 | 1163 | 1163 | 6538 | 1634 | 1633 | 108208 | 1204 | 1203 |
| CRP | 176422 | 3281 | 820 | 819 | 4581 | 1145 | 1144 | 75391 | 1034 | 1034 |
| Neutrophill # | 249945 | 4642 | 1160 | 1159 | 6515 | 1628 | 1627 | 62076 | 1202 | 1201 |
| MCV | 250359 | 4660 | 1164 | 1164 | 6541 | 1634 | 1634 | 108256 | 1204 | 1203 |
| ApoA | 161594 | 3029 | 756 | 756 | 4182 | 1045 | 1044 | - | - | - |
| Reticulocyte % | 246092 | 4531 | 1132 | 1131 | 6367 | 1591 | 1590 | - | - | - |
| ALP | 177485 | 3293 | 823 | 822 | 4611 | 1152 | 1152 | 105030 | 1034 | 1034 |
| LDL direct | 177383 | 3291 | 822 | 821 | 4603 | 1150 | 1150 | 72866 | 1034 | 1034 |
| Monocyte # | 249782 | 4639 | 1159 | 1158 | 6503 | 1625 | 1625 | 62076 | 1202 | 1202 |
| RDW | 249871 | 4645 | 1161 | 1160 | 6523 | 1630 | 1630 | - | - | - |
| Urea | 177343 | 3289 | 821 | 821 | 4603 | 1150 | 1149 | - | - | - |
| Triglycerides | 177317 | 3289 | 821 | 821 | 4603 | 1150 | 1149 | 105597 | 1034 | 1034 |
| Basophill % | 249368 | 4623 | 1155 | 1154 | 6495 | 1623 | 1622 | 62076 | 1199 | 1199 |
| Total protein | 162440 | 3045 | 761 | 760 | 4190 | 1047 | 1046 | 113509 | 1034 | 1034 |
| Urine sodium | 250253 | 4734 | 1183 | 1182 | 6497 | 1624 | 1623 | - | - | - |
| IGF-1 | 176643 | 3272 | 817 | 817 | 4583 | 1145 | 1144 | - | - | - |
| Eosinophill # | 249594 | 4629 | 1157 | 1156 | 6503 | 1625 | 1624 | 62076 | 1198 | 1197 |

African and South Asian GWASs used UK Biobank samples, Biobank Japan was used for East Asian GWAS summary data, for East Asians numbers are only shown for those traits with overlapping BBJ summary statistics. Across all traits 10,000 European samples were used as test data. Height - Standing height, BMI - Body mass index, CRP - C-reactive protein, MCV - Mean corpuscular volume, ApoA - Apolipoprotein A, Alp - Alkaline phosphatase, RDW - red cell distribution width.

**Extended Data Table 2 | *Bio*Me Biobank sample sizes for individuals of African, South Asian and East Asian ancestry**

|  | African | South Asian | East Asian |
|---|---|---|---|
| Standing height | 4061 | 560 | 589 |
| Body mass index | 4084 | 560 | 595 |
| LDL direct | 1998 | 217 | 220 |
| Mean corpuscular volume | 2412 | 360 | 366 |
| Platelet count | 2459 | 368 | 369 |
| Monocyte count | 2229 | 283 | 310 |
| Neutrophill count | 2226 | 284 | 309 |
| Eosinophill count | 2204 | 268 | 306 |
| Red blood cell distribution width | 2398 | 366 | – |

These samples were used for out-of-sample PRS validation.

# Reporting Summary

## Statistics

For all statistical analyses, confirm that the following items are present in the figure legend, table legend, main text, or Methods section.

| n/a | Confirmed | |
|---|---|---|
| ☐ | ☒ | The exact sample size (*n*) for each experimental group/condition, given as a discrete number and unit of measurement |
| ☐ | ☒ | A statement on whether measurements were taken from distinct samples or whether the same sample was measured repeatedly |
| ☐ | ☒ | The statistical test(s) used AND whether they are one- or two-sided<br>*Only common tests should be described solely by name; describe more complex techniques in the Methods section.* |
| ☐ | ☒ | A description of all covariates tested |
| ☒ | ☐ | A description of any assumptions or corrections, such as tests of normality and adjustment for multiple comparisons |
| ☐ | ☒ | A full description of the statistical parameters including central tendency (e.g. means) or other basic estimates (e.g. regression coefficient) AND variation (e.g. standard deviation) or associated estimates of uncertainty (e.g. confidence intervals) |
| ☐ | ☒ | For null hypothesis testing, the test statistic (e.g. *F*, *t*, *r*) with confidence intervals, effect sizes, degrees of freedom and *P* value noted<br>*Give P values as exact values whenever suitable.* |
| ☐ | ☒ | For Bayesian analysis, information on the choice of priors and Markov chain Monte Carlo settings |
| ☒ | ☐ | For hierarchical and complex designs, identification of the appropriate level for tests and full reporting of outcomes |
| ☐ | ☒ | Estimates of effect sizes (e.g. Cohen's *d*, Pearson's *r*), indicating how they were calculated |

*Our web collection on statistics for biologists contains articles on many of the points above.*

## Software and code

Policy information about availability of computer code

**Data collection**   *Provide a description of all commercial, open source and custom code used to collect the data in this study, specifying the version used OR state that no software was used.*

**Data analysis**   BridgePRS software, example data and tutorial for guiding its use are available from www.bridgeprs.net. Source code, to which www.bridgeprs.net links, is available from https://github.com/clivehoggart/BridgePRS, DOI badge https://zenodo.org/badge/latestdoi/452809505. Scripts used for all analyses are available on GitHub: https://github.com/clivehoggart/BridgePRS data
All other code
- PLINK v1.90: https://www.cog-genomics.org/plink/1.9/
- PLINK v2: https://www.cog-genomics.org/plink/2.0/
- LDSC version 1.01: https://github.com/bulik/ldsc
- METAL version 2011-03-25 - http://csg.sph.umich.edu/abecasis/metal/
- IMPUTE2 v2 https://mathgen.stats.ox.ac.uk/impute/impute_v2.html
- R version 4.0.3 https://cran.r-project.org
- Ridge regression - glmnet package (version 4.0-2) https://cran.r-project.org/web/packages/glmnet/index.html
- bootstrapping - boot package (version 1.3.25) https://cran.r-project.org/web/packages/boot/index.html
- PRS-CSx v1.0.0 (https://github.com/getian107/PRScsx)
- PRS-CS  v1.0.0 (https://github.com/getian107/PRScs)
- PRSice-2 v2 (https://www.prsice.info)
- HAPGEN v2.2.0 (01/04/2011):  https://mathgen.stats.ox.ac.uk/genetics_software/hapgen/hapgen2.htm

For manuscripts utilizing custom algorithms or software that are central to the research but not yet described in published literature, software must be made available to editors and reviewers. We strongly encourage code deposition in a community repository (e.g. GitHub). See the Nature Portfolio guidelines for submitting code & software for further information.

## Data

Policy information about availability of data

All manuscripts must include a data availability statement. This statement should provide the following information, where applicable:

- Accession codes, unique identifiers, or web links for publicly available datasets
- A description of any restrictions on data availability
- For clinical datasets or third party data, please ensure that the statement adheres to our policy

Publicly available data used to generate the simulated data are available from the following sites: 1000G Phase 3 reference panels: https://mathgen.stats.ox.ac.uk/ impute/1000GP Phase3. html and genetic maps for each subpopulation: ftp.1000genomes.ebi.ac.uk/vol1/ ftp/technical/working/20130507 omni recombination rates

UK Biobank genotype and phenotype data were obtained from the UK Biobank Resource under applica- tion 18177 https://www.ukbiobank.ac.uk/enable-your-research/approved-research/multi-trait-gwas-analyses-in-the-uk-biobank. UK Biobank Quality Control information (missingness, allele frequency, Hardy Weinberg Equilibrium) was obtained from UK Biobank resource 531: https://biobank.ctsu.ox.ac.uk/crystal/refer.cgi?id=531

Recruitment and enrollment of participants into the Mount Sinai BioMe Biobank is IRB and HIPAA approved. It is an electronic medical record-linked biobank which allows the use of de-identified samples linkable to past, present and future clinical information from electronic health records at Mount Sinai. Biome contains protected health information and is thus under controlled access. Application to access the data can be made to biome@mountsinai.org, also see https://icahn.mssm.edu/research/ipm/programs/biome-biobank.

BBJ summary statistics were downloaded from PheWeb: https://pheweb.jp.

SNP weights for the polygenic risk scores estimated by BridgePRS in this paper are available on Github https://github.com/clivehoggart/BridgePRS data

# Field-specific reporting

Please select the one below that is the best fit for your research. If you are not sure, read the appropriate sections before making your selection.

☒ Life sciences ☐ Behavioural & social sciences ☐ Ecological, evolutionary & environmental sciences

For a reference copy of the document with all sections, see nature.com/documents/nr-reporting-summary-flat.pdf

# Life sciences study design

All studies must disclose on these points even when the disclosure is negative.

| | |
|---|---|
| Sample size | Biobank analyses: We used all samples that were available for trait analyses in Biobanks, sample sizes for each phenotype in each ancestral population in UKB, BBJ and Biome are reported in Supplementary Tables 1 and 2.<br><br>Simulation analyses: Sample sizes for simulation studies were chosen as those typical for current GWASs: 80K European, 20K non-European. We show in the manuscript that there is a linear relationship between GWAS sample size and heritability, therefore, we performed simulation at 3 levels of heritability: 25%, 50% and 75%, the later two heritabilities are equivalent to doubling and trebling the sample size respectively with a heritability of 25%. To confirm the relationship between heritability and sample size analyses were run with half the sample size: 40K European, 10K non-European. |
| Data exclusions | Standard genotype quality controls were performed in both cohorts and are reported in the Methods. In UKB, samples with phenotypic values 6 standard deviation away from the mean were excluded. In Biome samples with phenotypic values 3 standard deviation away from the mean were excluded |
| Replication | Biobank analyses: PRS estimated using UKB and BBJ summary data were replicated in (1) unseen UKB samples and (2) the independent Mount Sinai Biome Biobank. Confidence intervals (CIs) for the R2 in the replication cohorts were calculated via boot strapping of 10,000 replicates, resulting CIs were symmetrically distributed around the mean values indicating consistency.<br>Simulation analyses: Each simulation setting was repeated 10 times. Results were consistent across replicates. |
| Randomization | UKB samples were randomly assigned to training, test and validation cohorts In each ancestral population. |
| Blinding | Researchers were blinded to group assignments. All phenotypes analyses were continuous, therefore phenotype blinding is not applicable. |

# Reporting for specific materials, systems and methods

We require information from authors about some types of materials, experimental systems and methods used in many studies. Here, indicate whether each material, system or method listed is relevant to your study. If you are not sure if a list item applies to your research, read the appropriate section before selecting a response.

## Materials & experimental systems

| n/a | Involved in the study |
|-----|----------------------|
| ☒ | ☐ Antibodies |
| ☒ | ☐ Eukaryotic cell lines |
| ☒ | ☐ Palaeontology and archaeology |
| ☒ | ☐ Animals and other organisms |
| ☐ | ☒ Human research participants |
| ☒ | ☐ Clinical data |
| ☒ | ☐ Dual use research of concern |

## Methods

| n/a | Involved in the study |
|-----|----------------------|
| ☒ | ☐ ChIP-seq |
| ☒ | ☐ Flow cytometry |
| ☒ | ☐ MRI-based neuroimaging |

## Human research participants

Policy information about studies involving human research participants

| | |
|---|---|
| Population characteristics | The UK Biobank is a population cohort. The Mount Sinai BioMe Biobank is a ~60k patient electronic medical record-linked biobank at Mount Sinai hospital and enables researchers to rapidly and efficiently conduct genetic, epidemiologic, molecular, and genomic studies on large collections of research specimens linked with medical information.<br>It has been enrolling since September 2007.<br>The ~60k participants represent diverse ancestry classified as African American ~20%, European American ~29%, East Asian (~4%) , South Asian (~3%), Hispanic (~36%) and OTHER (~9%) group |
| Recruitment | This research has been conducted using the UK Biobank Resource under Application Number 18177 (P.F.O'Reilly). The Mount Sinai Biome Biobank recruited patients visiting Mount Sinai hospital. |
| Ethics oversight | The study protocols were approved by the institutional review board at the Icahn School of Medicine at Mount Sinai. Participants from the UK Biobank provided written informed consent (Information available at https://www.ukbiobank.ac.uk/2018/02/gdpr/). All DNA samples and data in this study were pseudonymized. |

Note that full information on the approval of the study protocol must also be provided in the manuscript.

