## [Peer Review File · Nature Genetics]

Peer Review Information

Manuscript Title: BridgePRS leverages shared genetic effects across ancestries to increase polygenic risk score portability

Corresponding author name(s): Dr Clive Hoggart, Dr Paul O'Reilly

Reviewer Comments & Decisions:

Decision Letter, initial version:
--

24th Feb 2022

Dear Dr Hoggart,

Your Brief Communication entitled "BridgePRS: A powerful trans-ancestry Polygenic Risk Score method" has now been seen by 2 referees, whose comments are attached. In the light of their advice we have decided that we cannot offer to publish your manuscript in Nature Genetics.

While the referees find your work of some interest, they raise concerns about the strength of the novel conclusions that can be drawn at this stage. We feel that these reservations are sufficiently important as to preclude publication of this study in Nature Genetics.

Although we regret that we cannot offer to publish your paper in Nature Genetics given these reviews, I have discussed your manuscript and the reviewers' comments with our colleagues at Nature Communications. They would send the appropriately revised version back to the original referees if you transfer the revised manuscript to Nature Communications. Should you wish to have your revised paper considered by Nature Communications, please use the link to the Springer Nature manuscript transfer service in the footnote once the revision is ready, and include a point-by-point response to the reviewers' concerns.

Nature Communications ask that you respond to all the reviewer concerns and in particular that you address the comments regarding simulation studies, comparison with approaches other than PRS-CSx, and restriction of PRS-CSx to HapMap SNPs.

Your handling editor at Nature Communications would be Dr Tom Hearn (tom.hearn@springernature.com). If there is anything you would like to discuss before transferring the paper and its reviews, please don't hesitate to contact him by e-mail.

Please note that Nature Communications is a fully open access journal. For information about article processing charges, open access funding, and advice and support from Springer Nature, please

consult the Nature Communications Open Access page (www.nature.com/ncomms/open_access/index.html).

I am sorry that we cannot be more positive on this occasion but hope that you will find our referees' comments helpful when preparing your paper for submission elsewhere.

With all best wishes,
Wei

Wei Li, PhD
Senior Editor
Nature Genetics
New York, NY 10004, USA
www.nature.com/ng

Reviewers' Comments:

Reviewer #1:
Remarks to the Author:

The manuscript presents a new method to improve trans-ethnic polygenic prediction. The approach is relatively simple, sound and the derivations are correct. The comparative performance relative to the other similar method is convincing and exemplified for a decent number of traits. The paper is in head-to-head competition with a competing work by (Ruan et al), so my opinion is that the method of this (Hoggart et al) paper is better presented and smarter (more intuitive), the examples and analyses are, however, more thorough by Ruan et al. The advantage of this work is that it explored more traits, hence the results are more solid, while the competing work compared more methods (and larger cohorts), but for lesser traits. Below I provided detailed comments on how this current manuscript could be improved.

Major comments:

An overview figure would be very helpful to show how the different BridgePRS parameters [λ , α , τ , ψ , etc.] are chosen (via cross-validation) using the different training-test-validation sets.

A disadvantage of the method that as it requires a large number of cross-validations and if multiple stage 1 studies were available (from different populations), the weight of each study would need to be tested and grows exponentially with the number of studies. The application used only two populations to predict one trait in further samples in one of them. Could not have used all three populations to build a predictor (e.g. EUR+AFR+SAS to predict AFR)?

Speed et al (2021 Nat Gen) has implemented MAF-dependent genetic architecture, however accurate estimation of the "best" selection strength (α) estimation only becomes possible when the 64-parameter model is applied, i.e. when stratified for functional categories. I wonder how much sense it makes to fit this to a two-parameter model (heritability and selection strength)? Can the authors

report the alpha values and heritability estimates they got for the different traits? How do these compare to methods obtained by LDAK?

Mixed PRSs have been proposed in the past to improve transferability of PRSs to different populations (<https://www.ncbi.nlm.nih.gov/pmc/articles/PMC5726434/>) how do these approaches compare to BridgePRS?

How much information is lost by using hard-coded genotypes instead of continuous allele dosages?

While it is clear that the aim of this manuscript is to show that MAF-dependent genetic architecture can improve cross-ethnic predictions, but some follow-up analyses would have been insightful for the general audience of Nat Gen. E.g. other indices of performance than R^2 , do they obtain better stratified heritability (based on functional categories), etc. A richer Discussion on why BridgePRS out/under-performs in certain scenarios. An example of more extended discussions/analyses can be found in the competing paper [<https://www.medrxiv.org/content/10.1101/2020.12.27.20248738v1.full.pdf>].

I wondered that a more sophisticated model (e.g. BLD-LDAK) would work despite ignoring that the prediction population is not being the same as the training one? In general it would be very important to show much using population-ignorant PRSs perform (compared to BridgePRS) to better motivate this study (a more extensive comparison, such as here <https://pubmed.ncbi.nlm.nih.gov/34304866/>, <https://journals.plos.org/plosgenetics/article?id=10.1371/journal.pgen.1009021>) These previous work already highlighted candidates for strong methods (LDPred2, SBayesR), which would worth being applied naively (using EUR PRS to AFR predictions) to BridgePRS.

The authors state that the main advantage of their method over PRS-CSx is that the latter only uses HapMap SNPs. Why is it necessary for that method to use HapMap SNPs only? Those authors seem to have just made an arbitrary choice of using HapMap3 SNPs, but could have precalculated LD for a larger set of SNVs.

It is not clear how sensitive BridgePRS is to the choice of the LD matrix estimation: it seemed to me that the same cohort was used to estimate the LD structure in stage 1 and in the test set (when parameters are tuned) which provided the summary statistics. If so, would be important to check how robust is the method to estimating the LD matrices (Ψ) from a different set (e.g. 1000 Genomes)?

More information would be helpful on the optimally chosen parameters for each trait (P-value threshold, λ_0 , α , τ).

I miss simulation studies to show how performance of BridgePRS is expected to improve as a function of the magnitude of α , τ . Such simulation studies would help finding the reasons between performance and estimated parameters.

Minor comments:

1. Typos: "qgenotyped" -> "genotyped"; "the posterior model weights dor models" -> "the posterior model weights for models"; "the spareness of the genetic architecture" -> "the sparseness of the genetic architecture"

2. " $\beta \sim N(0, \psi * \lambda * I)$ ": I don't get the intuition why ψ (residual variance) is included here?
3. In the definition of λ_k (page 6), I assume that there is a typo and the authors wanted to write $\lambda_k = \lambda_0 * (\theta_k(1-\theta_k))^{-\alpha}$, i.e. the power must be negative to reflect stabilizing selection.
4. On page 5, the $Xt*y$ expression including $\theta*(1-\theta)*\hat{\beta}$ should be explained that those are all element-wise multiplications, unlike other multiplications with the same "notation". Also a factor of "2" is missing from that equation $Xt*y = 2*n*\theta*(1-\theta)*\hat{\beta}$
5. In Table 1, the signs of relative improvement (%) should be negative when BridgePRS has inferior performance.

Reviewer #2:

Remarks to the Author:

This paper describes a new approach (BridgePRS) to construct PRS under the cross-population setting. The idea behind is similar to the transfer learning, where in the first stage, the authors tune the parameters for PRS in a Bayesian framework using the large-scale population 1; and in the second stage, the authors plug in the optimal parameters from the first stage to another Bayesian framework and do parameter tuning using the population 2. Using this method, the authors constructed PRS across 19 traits for the African and South Asian people in UK Biobank and mainly compared their results with PRS-CSx. In addition, they also included an external validation analysis using the BioMe cohorts. Overall, it is nice to new methods are emerging to target this important topic – "cross-population genetic risk prediction"; however, as to the method and results in the current manuscript, I have a number of concerns.

1. One significant concern is from the comparison of BridgePRS and PRS-CSx. The authors claim that BridgePRS is superior to PRS-CSx, but the three observations below are against this claim:

- (a) In the summary table (Table1), when using imputed data, for the SAS population, BridgePRS is better than PRS-CSx in less than 50% of the 19 traits for both UKBB data and BioMe data (actually only 21% in UKBB).
- (b) When only using genotype data, the BridgePRS looks to be better than PRS-CSx, but this is probably because PRS-CSx used much fewer SNPs since it took the overlap between genotype data and HapMap variants; while in the real application, it is not clear to me if there are any reasons people would use genotype data instead of imputation data to make genetic predictions when using PRS-CSx.
- (c) In Figure 1, when comparing BridgePRS and PRS-CSx for each trait, we can hardly see statistically significant improvements of BridgePRS over PRS-CSx with overlapping confidence intervals for most traits in the analyses.

Taken together, the current major results are not persuasive enough to show the superiority of BridgePRS.

2. The simulation analyses are necessary and will be very helpful for the audience to better understand the pros and cons of the method: e.g. how the prediction performance of this method may be affected by sample size, heritability, genetic architecture, etc.

3. In the UKBB, there are also thousands of East Asian individuals, and in the paper of PRS-CSx, they also included the East Asian population in the analyses, are there any reasons that the current paper didn't include that?

4. Method - stage2: the authors used the pseudo-F-statistic to do the ranking and said this could help assign the same number of loci to each subset as the SNP P-value ranking. I'm curious how different will the assignment be if they keep using the SNP P-value ranking and add a restriction on the number of loci to each subset.
5. Method - Incomplete SNP overlap between populations 1 and 2. The authors conducted the stage-1 analyses and then took the intersections of variants to do the stage-2. Are there any reasons why the authors don't do the intersections of variants at the very beginning and then do these two-stage analyses? My intuition is that conducting the intersections after stage-1 will lead to unnecessary training in stage-1 and potential information loss when transferring to stage-2.
6. Method - Combining PRS. For model (3), I'm wondering if there might be any underlying motivation that may support that merging all candidate PRS could lead to better PRS other than simply having more parameters involved?
7. It will also be great to see the state-of-the-art single PRS methods and PRS-meta/PRS-multi in the comparison panel.

Decision Letter, Appeal:

10th Mar 2023

Dear Dr. Hoggart,

Thank you for your message of asking us to reconsider our decision on your manuscript "BridgePRS: A powerful trans-ancestry Polygenic Risk Score method". I have now discussed the points of your letter with my colleagues, and we think that you have some valid points. We therefore invite you to upload your revised manuscript and all necessary forms so that we can send your manuscript for peer review.

When preparing a revision, please ensure that it fully complies with our editorial requirements for format and style; details can be found in the Guide to Authors on our website (<http://www.nature.com/ng/>).

Please be sure that your manuscript is accompanied by a separate letter detailing the changes you have made and your response to the points raised.

****Please provide a full point-by-point response to reviewers.****

At this stage we will also need you to upload:

1) a copy of the manuscript in MS Word .docx format.

2) The Editorial Policy Checklist:

<https://www.nature.com/documents/nr-editorial-policy-checklist.pdf>

3) The Reporting Summary:

(Here you can read about the role of the Reporting Summary in reproducible science:

<https://www.nature.com/news/announcement-towards-greater-reproducibility-for-life-sciences->

research-in-nature-1.22062)

Please use the link below to be taken directly to the site and upload your files:

[redacted]

With kind wishes,

Wei Li, PhD
Senior Editor
Nature Genetics
New York, NY 10004, USA
www.nature.com/ng

Author Rebuttal to Initial comments

Reviewer #1:

Remarks to the Author:

The manuscript presents a new method to improve trans-ethnic polygenic prediction. The approach is relatively simple, sound and the derivations are correct. The comparative performance relative to the other similar method is convincing and exemplified for a decent number of traits. The paper is in head-to-head competition with a competing work by (Ruan et al), so my opinion is that the method of this (Hoggart et al) paper is better presented and smarter (more intuitive), the examples and analyses are, however, more thorough by Ruan et al. The advantage of this work is that it explored more traits, hence the results are more solid, while the competing work compared more methods (and larger cohorts), but for lesser traits. Below I provided detailed comments on how this current manuscript could be improved.

Major comments:

An overview figure would be very helpful to show how the different BridgePRS parameters [λ , α , τ , ψ , etc.] are chosen (via cross-validation) using the different training-test-validation sets.

Overview figure of the method is now included

A disadvantage of the method that as it requires a large number of cross-validations and if multiple stage 1 studies were available (from different populations), the weight of each study would need to be tested and grows exponentially with the number of studies. The application used only two populations to predict one trait in further samples in one of them. Could not have used all three populations to build a predictor (e.g. EUR+AFR+SAS to predict AFR)?

We now present results using BBJ + AFR(ukb) + EUR(ukb) summary stats to predict into UKB African samples for Bridge. These analyses show marginal increases in predictive accuracy compared to using AFR(ukb) + EUR(ukb) for height, BMI and monocyte count, but no improvement in the other traits. We note that use of two populations will likely be the most typical scenario in most studies given the predominance of European GWAS data.

Speed et al (2021 Nat Gen) has implemented MAF-dependent genetic architecture, however accurate estimation of the “best” selection strength (alpha) estimation only becomes possible when the 64-parameter model is applied, i.e. when stratified for functional categories. I wonder how much sense it makes to fit this to a two-parameter model (heritability and selection strength)? Can the authors report the alpha values and heritability estimates they got for the different traits? How do these compare to methods obtained by LDAK?

We agree that there is large uncertainty associated with alpha. Therefore, rather than choose a single “best guess” of alpha, we account for the uncertainty by weighting models with different alphas. While our method could estimate and output mean alphas and also heritability (although this is not a parameter in the present model) it has not been developed to optimise these and thus this functionality has not been included here especially given other bespoke methods like GCTA and LDAK exist for doing this. We also note that the leading competing method, PRS-CSx, now published in Nat Gen, only provides updated effect size estimates for each ancestry GWAS and does not perform any of the downstream PRS analyses incorporated into BridgePRS.

Mixed PRSs have been proposed in the past to improve transferability of PRSs to different populations (<https://www.ncbi.nlm.nih.gov/pmc/articles/PMC5726434/>) how do these approaches compare to BridgePRS?

We now compare BridgePRS to the weighted multi-ancestry method described in the cited paper using PRS-CS single ancestry PRS which Ruan et al found to be the better performing of C+T and LDpred2 in this application. We have also applied C+T to the meta-analysis of the contributing GWAS summary statistics, using the reference population LD panel that which maximises prediction in the test data, this approach was also implemented in Ruan et al.

How much information is lost by using hard-coded genotypes instead of continuous allele dosages?

This has been explored by others in great depth, see for example Kutalik et al (2011) and Ding et al (2022), and is beyond the scope of this paper. However, our method is not limited to hard coded genotypes and can equally make predictions and calculate LD using dosage data in pgen format.

While it is clear that the aim of this manuscript is to show that MAF-dependent genetic architecture can improve cross-ethnic predictions, but some follow-up analyses would have been insightful for the general audience of Nat Gen. E.g. other indices of performance than R^2 , do they obtain better stratified heritability (based on functional categories), etc. A richer Discussion on why BridgePRS out/under-performs in certain scenarios. An example of more extended discussions/analyses can be found in the competing paper

[\[https://www.medrxiv.org/content/10.1101/2020.12.27.20248738v1.full.pdf\]](https://www.medrxiv.org/content/10.1101/2020.12.27.20248738v1.full.pdf).

We have now performed extensive simulations that have revealed the genetic architectures and scenarios for which each method is optimal.

I wondered that a more sophisticated model (e.g. BLD-LDAK) would work despite ignoring that the prediction population is not being the same as the training one? In general it would be very important to show much using population-ignorant PRSs perform (compared to BridgePRS) to better motivate this study (a more extensive comparison, such as here <https://pubmed.ncbi.nlm.nih.gov/34304866/>, <https://journals.plos.org/plosgenetics/article?id=10.1371/journal.pgen.1009021>) These previous work already highlighted candidates for strong methods (LDPred2, SBayesR), which would worth being applied naively (using EUR PRS to AFR predictions) to BridgePRS.

Ruan et al have shown that single population PRS are inferior to straightforward multi-ancestry PRS methods, therefore we feel there is little to be gained in replicating the full spectrum of this work, especially since we find that BridgePRS outperforms PRS-CSx in many scenarios. However, we now compare BridgePRS and PRS-CSx with two single ancestry PRS methods adapted to utilise trans-ancestry GWAS data, which were also considered in Ruan et al, (1) CS applied to two populations for which summary statistics are available and then optimally combined using a target population with individual level data (test set) (Ruan et al found CS to be the best of the single ancestry methods in this application) and (2) meta-analysis of the two summary statistics followed by clumping and thresholding using LD panel selected using the test set.

The authors state that the main advantage of their method over PRS-CSx is that the latter only uses HapMap SNPs. Why is it necessary for that method to use HapMap SNPs only? Those authors seem to have just made an arbitrary choice of using HapMap3 SNPs, but could have precalculated LD for a larger set of SNVs.

Analysing an unrestricted SNP set is not the only advantage of BridgePRS as demonstrated by our simulation which used only HapMap SNPs for all analyses. These highlight where each method is better and real data analyses showing overall greater performance for BridgePRS in populations of African descent. The reviewer is correct that PRS-Csx could theoretically be

applied to any SNP set, but in practice analyses are limited to HapMap SNPs with the current implementation and it would be beyond the expertise of the vast majority of users to generate LD panels for alternative SNP sets. We also note that the LD panels for HapMap variants requires 30Gb of storage, which is a greater problem for researchers in under-resourced regions of the world in which these methods may have greatest uptake.

It is not clear how sensitive BridgePRS is to the choice of the LD matrix estimation: it seemed to me that the same cohort was used to estimate the LD structure in stage 1 and in the test set (when parameters are tuned) which provided the summary statistics. If so, would be important to check how robust is the method to estimating the LD matrices (Ψ) from a different set (e.g. 1000 Genomes)?

Simulated data, generated from 1000G LD reference panel, are analysed using both UKB and 1000G data to estimate LD. Both BridgePRS and PRS-CSx have similar inferior prediction when using the UKB data as LD reference.

More information would be helpful on the optimally chosen parameters for each trait (P-value threshold, λ_0 , α , τ).

BridgePRS does not choose single best parameters but instead averages across models estimated across a spectrum of parameters within a ridge regression framework

I miss simulation studies to show how performance of BridgePRS is expected to improve as a function of the magnitude of α , τ . Such simulation studies would help finding the reasons between performance and estimated parameters.

Analyses of simulated data are now included

Minor comments:

1. Typos: “qgenotyped” -> “genotyped”; “the posterior model weights dor models” -> “the posterior model weights for models”; “the spareness of the genetic architecture” -> “the sparseness of the genetic architecture”

Corrected, thanks

2. “ $\beta \sim N(0, \psi * \lambda_k)$ ”: I don’t get the intuition why ψ (residual variance) is included here?

Lambda is included for conjugacy, this is now explained in the text

3. In the definition of λ_k (page 6), I assume that there is a typo and the authors

wanted to write $\lambda_k = \lambda_0 (\theta_k (1 - \theta_k))^{-\alpha}$, i.e. the power must be negative to reflect stabilizing selection.

Lambda is precision = 1 / variance, therefore power is positive

4. On page 5, the $X^T y$ expression including $\theta^*(1-\theta)*\hat{\beta}$ should be explained that those are all element-wise multiplications, unlike other multiplications with the same “notation”. Also a factor of “2” is missing from that equation $X^T y = 2*n*\theta*(1-\theta)*\hat{\beta}$

Corrected, thanks

5. In Table 1, the signs of relative improvement (%) should be negative when BridgePRS has inferior performance.

Information now shown as figures

Reviewer #2:

Remarks to the Author:

This paper describes a new approach (BridgePRS) to construct PRS under the cross-population setting. The idea behind is similar to the transfer learning, where in the first stage, the authors tune the parameters for PRS in a Bayesian framework using the large-scale population 1; and in the second stage, the authors plug in the optimal parameters from the first stage to another Bayesian framework and do parameter tuning using the population 2. Using this method, the authors constructed PRS across 19 traits for the African and South Asian people in UK Biobank and mainly compared their results with PRS-CSx. In addition, they also included an external validation analysis using the BioMe cohorts. Overall, it is nice to see new methods are emerging to target this important topic – “cross-population genetic risk prediction”; however, as to the method and results in the current manuscript, I have a number of concerns.

1. One significant concern is from the comparison of BridgePRS and PRS-CSx. The authors claim that BridgePRS is superior to PRS-CSx, but the three observations below are against this claim:

(a) In the summary table (Table1), when using imputed data, for the SAS population, BridgePRS is better than PRS-CSx in less than 50% of the 19 traits for both UKBB data and BioMe data (actually only 21% in UKBB).

(b) When only using genotype data, the BridgePRS looks to be better than PRS-CSx, but this is probably because PRS-CSx used much fewer SNPs since it took the overlap between genotype data and HapMap variants; while in the real application, it is not clear to me if there are any reasons people would use genotype data instead of imputation data to make genetic predictions when using PRS-CSx.

(c) In Figure 1, when comparing BridgePRS and PRS-CSx for each trait, we can hardly

see statistically significant improvements of BridgePRS over PRS-CSx with overlapping confidence intervals for most traits in the analyses.

Taken together, the current major results are not persuasive enough to show the superiority of BridgePRS.

Results are now shown for imputed data only, for these analyses we report p-values comparing average R^2 across traits between methods for each population in each cohort. These show that BridgePRS is superior to PRS-Csx for predicting trait values of African individuals, for South Asians the methods appear indistinguishable and for East Asians PRS-CSx is superior. These results are consistent with the extensive simulations we have now performed which demonstrate when and why each method is superior.

2. The simulation analyses are necessary and will be very helpful for the audience to better understand the pros and cons of the method: e.g. how the prediction performance of this method may be affected by sample size, heritability, genetic architecture, etc.

Analyses of simulated data now included, see above.

3. In the UKBB, there are also thousands of East Asian individuals, and in the paper of PRS-CSx, they also included the East Asian population in the analyses, are there any reasons that the current paper didn't include that?

There are only approximately 2,500 EAS samples in UKB across the phenotypes analysed, too few to split into GWAS samples and independent test and validation sets for meaningful inference. However, we have now used BBJ summary stats, which we combine with UKB EUR summary stats, to predict into East Asians in UKB and BioMe.

4. Method - stage2: the authors used the pseudo-F-statistic to do the ranking and said this could help assign the same number of loci to each subset as the SNP P-value ranking. I'm curious how different will the assignment be if they keep using the SNP P-value ranking and add a restriction on the number of loci to each subset.

Ranking loci using the pseudo-F-statistic gives marginal improvement over Eur SNP p-value ranking for prediction into UKB AFR and EAS samples. These analyses used imputed data, analyses using only genotyped SNPs show a more marked improvement using the pseudo-F-statistic ranking. Since the pseudo F-statistic does not produce inferior results and sometimes produces superior results we have retained this modelling of BridgePRS in the results presented in the paper. However, the BridgePRS script allows users to rank loci in stage 2 using either ranking method. Comparative results of pseudo F-statistic and p-value ranking are now described in the manuscript

5. Method - Incomplete SNP overlap between populations 1 and 2. The authors

conducted the stage-1 analyses and then took the intersections of variants to do the stage-2. Are there any reasons why the authors don't do the intersections of variants at the very beginning and then do these two-stage analyses? My intuition is that conducting the intersections after stage-1 will lead to unnecessary training in stage-1 and potential information loss when transferring to stage-2.

SNP overlap is taken in stage 2 to allow models fit in stage 1 to be applied to other data sets with different SNPs. This was important in our applications where models fit using European UKB samples were subsequently applied to African and South Asian samples with different SNP sets passing QC and the BBJ data. Incomplete SNP overlap in stage 2 is accounted in the multivariate normal prior distributions (estimated in stage 1) by recalculating the prior conditional on SNP effects of the non-overlapping variants set to zero. This is now described in the manuscript

6. Method – Combining PRS. For model (3), I'm wondering if there might be any underlying motivation that may support that merging all candidate PRS could lead to better PRS other than simply having more parameters involved?

The model could be further parameterised to incorporate hyperpriors to allow Bayesian learning of the prior parameters. However, we believe that merging PRS via shrinkage regression (ridge regression) gives a pragmatic and computationally efficient solution which avoids the otherwise necessary computational expense of MCMC

7. It will also be great to see the state-of-the-art single PRS methods and PRS-meta/PRS-multi in the comparison panel.

PRS-meta/PRS-multi are now included in all comparisons as implemented in the PRS-CSx paper. The published PRS-CSx paper demonstrated that PRS-CS was the best performing of the single ancestry PRS-multi implementations and therefore use this method alone as we feel there is little to be gained in replicating this work and applying the more complete spectrum of PRS-multi implementations.

Decision Letter, first revision:

30th May 2023

Dear Dr Hoggart,

Your Brief Communication, "BridgePRS: A powerful trans-ancestry Polygenic Risk Score method" has now been seen by 2 referees. You will see from their comments below that while they find your work

of interest, some important points are raised by Reviewer #2. We are interested in the possibility of publishing your study in Nature Genetics, but would like to consider your response to these concerns in the form of a revised manuscript before we make a final decision on publication.

We therefore invite you to revise your manuscript taking into account all reviewer and editor comments. Please highlight all changes in the manuscript text file. At this stage we will need you to upload a copy of the manuscript in MS Word .docx or similar editable format.

*2) If you have not done so already please begin to revise your manuscript so that it conforms to our Brief Communication format instructions, available [here](http://www.nature.com/ng/authors/article_types/index.html). Refer also to any guidelines provided in this letter.

[redacted]

We hope to receive your revised manuscript within four to eight weeks. If you cannot send it within this time, please let us know.

Nature Genetics is committed to improving transparency in authorship. As part of our efforts in this direction, we are now requesting that all authors identified as 'corresponding author' on published

papers create and link their Open Researcher and Contributor Identifier (ORCID) with their account on the Manuscript Tracking System (MTS), prior to acceptance. ORCID helps the scientific community achieve unambiguous attribution of all scholarly contributions. You can create and link your ORCID from the home page of the MTS by clicking on 'Modify my Springer Nature account'. For more information please visit www.springernature.com/orcid.

Sincerely,
Wei

Wei Li, PhD
Senior Editor
Nature Genetics
New York, NY 10004, USA
www.nature.com/ng

Reviewers' Comments:

Reviewer #1:

Remarks to the Author:

The authors have adequately addressed all my comments, I have no further comments/questions and recommend publication in its present form.

Reviewer #2:

Remarks to the Author:

The manuscript has been significantly improved through (1) extended simulation and real data analysis, and (2) a clearer statement of when the BridgePRS outperforms PRS-CSx. Thank the authors for making these improvements!

The authors have addressed most of my concerns; however, one major concern remains regarding the simulation study. The analysis results indicate that BridgePRS is superior to PRS-CSx only in the AFR population. The authors explain that this may be due to the small sample size of the AFR GWAS. In the discussion section, the authors summarize that "BridgePRS has higher performance relative to PRS-CSx..., for lower GWAS sample sizes." However, there is no such simulation scenario that considers different sample sizes for the same target population, which is essential to verify the authors' claim. Given that the major highlight of this work is that BridgePRS shows better performance than PRS-CSx in the AFR population (Fig 3d), it is worth adding the simulations to discuss this thoroughly.

Author Rebuttal, first revision:

Reviewers' Comments:

Reviewer #1:

Remarks to the Author:

The authors have adequately addressed all my comments, I have no further comments/questions and recommend publication in its present form.

We thank the reviewer for their original review, which helped us to improve our manuscript substantially. We are grateful that the reviewer recommends publication of the manuscript in its present form.

Reviewer #2:

Remarks to the Author:

The manuscript has been significantly improved through (1) extended simulation and real data analysis, and (2) a clearer statement of when the BridgePRS outperforms PRS-CSx. Thank the authors for making these improvements! The authors have addressed most of my concerns; however, one major concern remains regarding the simulation study. The analysis results indicate that BridgePRS is superior to PRS-CSx only in the AFR population. The authors explain that this may be due to the small sample size of the AFR GWAS. In the discussion section, the authors summarize that "BridgePRS has higher performance relative to PRS-CSx..., for lower GWAS sample sizes." However, there is no such simulation scenario that considers different sample sizes for the same target population, which is essential to verify the authors' claim. Given that the major highlight of this work is that BridgePRS shows better performance than PRS-CSx in the AFR population (Fig 3d), it is worth adding the simulations to discuss this thoroughly.

We thank the reviewer for their feedback about the significant improvement of our manuscript provided by our extended simulations, extended real data analysis, and our exposition of when BridgePRS outperforms PRS-CSx. We also thank the reviewer for highlighting that an important aspect of our method comparisons under simulation was not adequately described in the manuscript. In the previous submission, we included simulations at SNP heritabilities of 25% and 50% and we noted (Line 155) that: "Power of GWAS, and thus PRS, is a function of heritability and sample size, such that doubling heritability is equivalent to doubling sample size in terms of power"

However, we have now expanded on this substantially to ensure that this concern is addressed in multiple ways. We have:

- 1. Repeated our entire simulation study (presented as results in Fig.2) at half the GWAS sample size and have added the new results as Supplementary Figure 2.**
- 2. Justified the original statement linking sample size and heritability, adding a mathematical proof to the Supplementary Material.**
- 3. Added references demonstrating that this relationship holds for PRS, and that generalises the relationship to include the impact of varying number of causal variants.**
- 4. Performed an additional set of simulations to perform the methods benchmarking at a higher heritability, to expand the range of scenarios considered even further.**
- 5. Added additional text to the Results section highlighting the consistency of the simulation results with the described relationship. The new text is below:**

Expansion of original sentence (Line 158):

“Power of GWAS, and therefore PRS, is a function of sample size and heritability, such that doubling heritability is equivalent to doubling sample size in terms of power since the standard error of a GWAS regression coefficient is the same if either the sample size or heritability is doubled (see Supplementary Material)”

New text in the Results section (Line 176):

“The theoretical proportion of heritability captured by a PRS derived by C+T, assuming independent causal variants, is $r^2/h^2=(1+m/nh^2)^{-1}$, where r^2 is the variance explained by the PRS, m is the number of causal variants and n is the GWAS sample size [17, 18]. While BridgePRS and PRS-CSx are more sophisticated methods than C+T, able to model allelic heterogeneity and tackle winner's curse, the factor nh^2/m in the equation, which is a measure of power to detect individual causal variant effects, is useful in describing the relative performance of the methods. Fig. 2 shows results in relation to nh^2/m (up to a proportionality constant): lower values favour BridgePRS, higher values favour PRS-CSx, and within the same target population the relative performance of the methods is similar for constant nh^2/m . For example, results at 25% heritability and 5% causal variants show the same relative method performance as results at 50% heritability and 10% causal variants, for both African (Fig. 2a vs Fig. 2c) and East Asian (Fig. 2b vs Fig. 2d) target samples, as expected.”

“Supplementary Fig. 2 shows results for the same simulation settings used in the main analysis (Fig. 2) but with the GWAS training sample size halved (40K European, 10K non-European). Here we see that the performance of BridgePRS relative to PRS-CSx increases compared to results with the full GWAS samples sizes and, as predicted, the relative performance of the methods at 50% heritability is similar to that at 25% heritability and the full GWAS sample sizes. Supplementary Fig. 3 shows results at the original GWAS sample size and 75% heritability (equivalent to 240K European, 60K non-European GWAS training sample sizes and 25% heritability). As predicted, the performance of BridgePRS relative to PRS-CSx decreases compared to results at 25% and 50% heritability.”

Decision Letter, second revision:

15th Aug 2023

Dear Dr. Hoggart,

Thank you for submitting your revised manuscript "BridgePRS: A powerful trans-ancestry Polygenic Risk Score method" (NG-TR59296R2). It has now been seen by the original referees and their comments are below. The reviewers find that the paper has improved in revision, and therefore we'll be happy in principle to publish it in Nature Genetics, pending minor revisions to comply with our editorial and formatting guidelines.

Sincerely,
Wei

Wei Li, PhD
Senior Editor
Nature Genetics
www.nature.com/ng

Reviewer #2 (Remarks to the Author):

The authors have addressed all my comments, I have no further comments/questions.

Final Decision Letter:

20th Oct 2023

Dear Dr. Hoggart,

I am delighted to say that your manuscript "BridgePRS leverages shared genetic effects across ancestries to increase polygenic risk score portability" has been accepted for publication in an upcoming issue of Nature Genetics.

Your paper will be published online after we receive your corrections and will appear in print in the

next available issue. You can find out your date of online publication by contacting the Nature Press Office (press@nature.com) after sending your e-proof corrections. Now is the time to inform your Public Relations or Press Office about your paper, as they might be interested in promoting its publication. This will allow them time to prepare an accurate and satisfactory press release. Include your manuscript tracking number (NG-TR59296R3) and the name of the journal, which they will need when they contact our Press Office.

Please note that *Nature Genetics* is a Transformative Journal (TJ). Authors may publish their research with us through the traditional subscription access route or make their paper immediately open access through payment of an article-processing charge (APC). Authors will not be required to make a final decision about access to their article until it has been accepted. [Find out more about Transformative Journals](https://www.springernature.com/gp/open-research/transformative-journals)

Authors may need to take specific actions to achieve [compliance with funder and institutional open access mandates](https://www.springernature.com/gp/open-research/funding/policy-compliance-faqs). If your research is supported by a funder that requires immediate open access (e.g. according to [Plan S principles](https://www.springernature.com/gp/open-research/plan-s-compliance)) then you should select the gold OA route, and we will direct you to the compliant route where possible. For authors selecting the subscription publication route, the journal's standard licensing terms will need to be accepted, including [self-archiving and license to publish](https://www.nature.com/nature-portfolio/editorial-policies/self-archiving-and-license-to-publish). Those licensing terms will supersede any other terms that the author or any third party may assert apply to any version of the manuscript.

You can now use a single sign-on for all your accounts, view the status of all your manuscript submissions and reviews, access usage statistics for your published articles and download a record of

your refereeing activity for the Nature journals.

If you have not already done so, we invite you to upload the step-by-step protocols used in this manuscript to the Protocols Exchange, part of our on-line web resource, natureprotocols.com. If you complete the upload by the time you receive your manuscript proofs, we can insert links in your article that lead directly to the protocol details. Your protocol will be made freely available upon publication of your paper. By participating in natureprotocols.com, you are enabling researchers to more readily reproduce or adapt the methodology you use. [Natureprotocols.com](http://natureprotocols.com) is fully searchable, providing your protocols and paper with increased utility and visibility. Please submit your protocol to <https://protocolexchange.researchsquare.com/>. After entering your nature.com username and password you will need to enter your manuscript number (NG-TR59296R3). Further information can be found at <https://www.nature.com/nature-portfolio/editorial-policies/reporting-standards#protocols>

Sincerely,
Wei

Wei Li, PhD
Senior Editor
Nature Genetics
New York, NY 10004, USA
www.nature.com/ng